# Fair Text-to-Image Diffusion via Fair Mapping

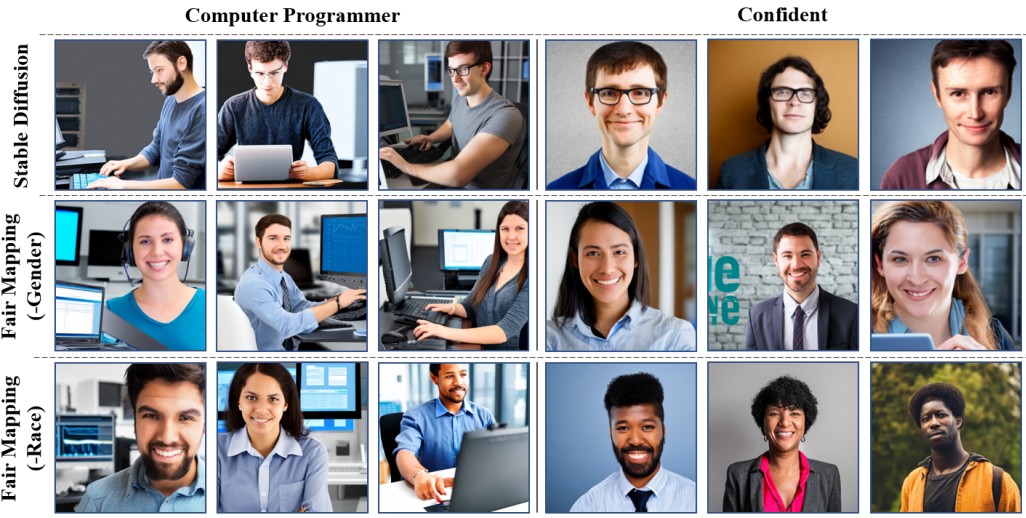

Figure 1: **Fair Diffusion Balancing Demographic Representation in Text-to-Image Models.** Our approach minimally adjusts parameters during training to eliminate demographic biases in pre-trained text-to-image models, resulting in more equitable image generation. Here, Stable Diffusion (top row) runs the risk of lacking diversity in its output, e.g., only male-appearing persons generation as *computer programmer* and *confident*). In contrast, Fair Diffusion (our method) allows the creation of a more equitable and unbiased representation of visual content.

## Abstract

In this paper, we address the limitations of existing text-to-image diffusion models in generating demographically fair results when given human-related descriptions. These models often struggle to disentangle the target language context from sociocultural biases, resulting in biased image generation. To overcome this challenge, we propose Fair Mapping, a general, model-agnostic, and lightweight approach that modifies a pre-trained text-to-image model by controlling the prompt to achieve fair image generation. One key advantage of our approach is its high efficiency. The training process only requires updating a small number of parameters in an additional linear mapping network. This not only reduces the computational cost but also accelerates the optimization process. We first demonstrate the issue of bias in generated results caused by language biases in text-guided diffusion models. By developing a mapping network that projects language embeddings into an unbiased space, we enable the generation of relatively balanced demographic results based on a keyword specified in the prompt. With comprehensive experiments on face image generation, we show that our method significantly improves image generation performance when prompted with descriptions related to human faces. By effectively addressing the issue of bias, we produce more fair and diverse image outputs. This work contributes to the field of text-to-image generation by enhancing the ability to generate images that accurately reflect the intended demographic characteristics specified in the text.

# 1 INTRODUCTION

Diffusion models (Ho et al., 2020; Song et al., 2021) have achieved remarkable performance in various applications, including image synthesis, video generation, and molecule design (Dhariwal & Nichol, 2021; Wu et al., 2022; Ceylan et al., 2023; Blattmann et al., 2023; Poole et al., 2023; Tevet et al., 2023). This innovation involves the deliberate incorporation of conditional elements (Ho & Salimans, 2022) into the iterative diffusion process, enabling a new dimension of control and flexibility in generative modelling. However, as diffusion models are increasingly used in real-world applications, addressing bias (Ouyang et al., 2022; Friedrich et al., 2023) becomes crucial as exemplified in Figure 1. Especially, for human-related description, fair face generation with diversity across demographic groups is essential to avoid perpetuating inequalities and biases (Ning et al., 2023; Kärkkäinen & Joo, 2021; Schramowski et al., 2023; Bansal et al., 2022).

Text-guided diffusion models tend to rely on linguistic bias, limiting their comprehensive understanding of visual and textual dimensions (Dehouche, 2021; Wang et al., 2022). When training on a dataset that includes images and accompanying texts (Schuhmann et al., 2022; 2021), text-generated diffusion models face challenges in achieving robust generalization. This challenge emerges when the model strongly associates biased contextual information present in both images and text prompts within the training data (Goyal et al., 2019). As illustrated in Figure 1, images generated by "An image of a computer programmer" and "An image of a confident person" predominantly depict male-related individuals, even if there is no explicit biased information in our input prompts. Moreover, as shown in Figure 2, both linguistic bias and generated outputs in diffusion models exhibit inclinations towards males, indicating a tendency to generate male-associated content (Gaucher et al., 2011; Friedrich et al., 2023) (see Section B in Appendix for details). In fact, such a phenomenon is ubiquitous for variants of text-to-image diffusion models (see Figure 4 for other diffusion models), especially when we want to generate human face-related images. These bias issues in the diffusion model can arise from various factors, including training data, system design, and parameter settings.

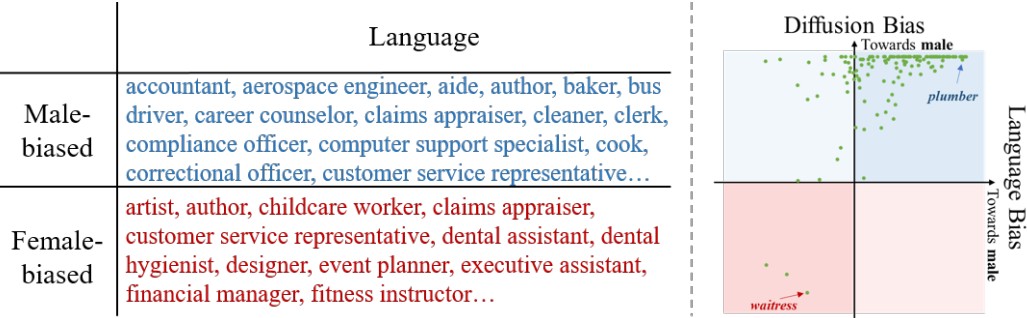

Figure 2: **Language Bias and Generative Bias Visualization.** We conduct a bias analysis of the language characteristics and the generated outcomes during the diffusion process. Left: Examples of language prejudice. Right: Linguistic bias and diffusion generative bias in occupational data. Each point represents a profession.

Addressing language bias in text-guided diffusion models can involve enhancing training data through annotations or data augmentation (Wang et al., 2022; Dehouche, 2021). However, building a large-scale, high-quality dataset that ensures equitable representation across diverse populations is a challenging task in practice (Schuhmann et al., 2022; Struppek et al., 2022). Real-world data is often biased and incomplete, reflecting inherent stereotypes in human perceptions (Makady et al., 2017; Bansal et al., 2022). Consequently, the pressing question remains: *How can unbiased inferences be made in the face of training data that is inherently biased?* While some efforts have started to address fairness concerns in generating face images using diffusion models, such as the study by Friedrich et al. (2023), shifting bias by human instruction. However, these endeavors are limited in manually modifying, lacking a robust framework for measuring bias in outcomes, which undermines trust and confidence in the fairness of the results. This highlights the need for comprehensive strategies to tackle the overarching challenge of fairness in diffusion models (Zhang et al., 2023).

In this paper, we conduct a comprehensive investigation into the influence of linguistic biases within text-guided diffusion models and their subsequent impact on the generated outcomes. We propose a novel post-processing, model-agnostic, and lightweight method namely **Fair Mapping**. Briefly

speaking, there are two additional components compared to vanilla diffusion models: The first one is a linear mapping network which is strategically designed to rectify the implicit bias in representation vectors given by the text encoder in text-to-image diffusion models. It addresses the disentanglement of the target language context from additional language biases by introducing a fair penalty mechanism. This mechanism fosters a harmonious representation of sensitive information within word embeddings via a linear network with only a modest addition of new parameters. At the inference stage, Fair Mapping introduces a detector to identify whether the user's input prompt contains implicit target information and explicit sensitive attributes on its expected generated images.

This paper makes several significant contributions and highlights key findings: 1. *In-depth Analysis of Bias in text-guided diffusion models:* Our work comprehensively explores and explains the bias issue in generated results caused by language biases within text-guided diffusion models, shedding light on the contributing dynamics. 2. *Innovative Fair Mapping Module:* Our novel fair mapping module optimizes minimal parameters for training, enabling seamless integration into classifier-free guided text-based generative models. Importantly, it achieves fairness in generative outputs without modifying the model's original structure, presenting a significant advancement in the field. 3. *Proposed Fairness Evaluation Metric:* Alongside our innovative fair mapping module, we introduce the **first** evaluation metric designed to assess the fairness of diffusion models in generating text-guided human-related images. This metric provides a systematic and objective measure for quantifying the reduction of bias in the generative process, enabling more precise evaluation of fairness outcomes.

## 2 RELATED WORK

**Text-guided Diffusion Models:** Text-guided diffusion models merge textual descriptions with visual content to create high-resolution, realistic images that align with the semantic guidance provided by the accompanying text prompts (Ramesh et al., 2022; Saharia et al., 2022; El-Nouby et al., 2018; Kim et al., 2023; Avrahami et al., 2023; Balaji et al., 2022; Feng et al., 2023b; He et al., 2023). However, this fusion of modalities also brings to the forefront issues related to bias and fairness (Struppek et al., 2022; Bansal et al., 2022), which have prompted extensive research efforts to ensure that the generated outputs do not perpetuate societal inequalities or reinforce existing biases. In this paper, we delve into these challenges and the state-of-the-art solutions aimed at enhancing the fairness and equity of text-guided diffusion models.

**Fairness and Bias in Diffusion Models:** Fair data generation is crucial for generative modelling to ensure discrimination-free and unbiased data. While large internet datasets are commonly used in data-driven generative models, they often contain biased and degenerate human behavior (Birhane et al., 2021; Schuhmann et al., 2022; 2021). Existing research on widely bias and discrimination in generative models is limited. Notably, Xu et al. (2018) proposed fairness-aware generative adversarial networks to generate fair data with high utility, while Schramowski et al. (2023) developed a test platform to evaluate and mitigate undesired effects from unfiltered and imbalanced training datasets. Friedrich et al. (2023); Muñoz et al. (2023) emphasize post-processing methods, adjusting outputs after deployment to mitigate bias by human instruction. However, these approaches primarily address manipulation, overlooking the role of language representation in bias mitigation. This paper seeks to bridge this gap by addressing linguistic biases within text-guided diffusion models, thereby contributing to a more comprehensive understanding and mitigation of biases in generative data.

**Bias in Language Models:** The issue of bias in language models (LMs) has raised concerns about their potential to generate biased, racist, sexist, or toxic language. Ensuring fairness in these models has been extensively studied and validated, especially in the context of large-scale models (Gehman et al., 2020; Abid et al., 2021; Bender et al., 2021; Zhang et al., 2022; Radford et al., 2021; Tian et al., 2018; Ding et al., 2022). Efforts have been made to address these biases, with approaches introduced by Wang et al. (2022); Dehouche (2021) aiming to mitigate the impact of bias. In our work, we explore the intersection of bias mitigation efforts in both language models and generative models, which is a critical juncture in the pursuit of fairness and ethics in artificial intelligence.

## 3 METHOD

In this section, we introduce our Fair Diffusion model framework, outlining its key components and novel features designed to address biases in text-to-image generation. Our objective is to improve the ability of text-to-image models to produce equitable results when generating faces from human-related descriptions. This enhancement is achieved through small fraction parameter optimizations during the training process.

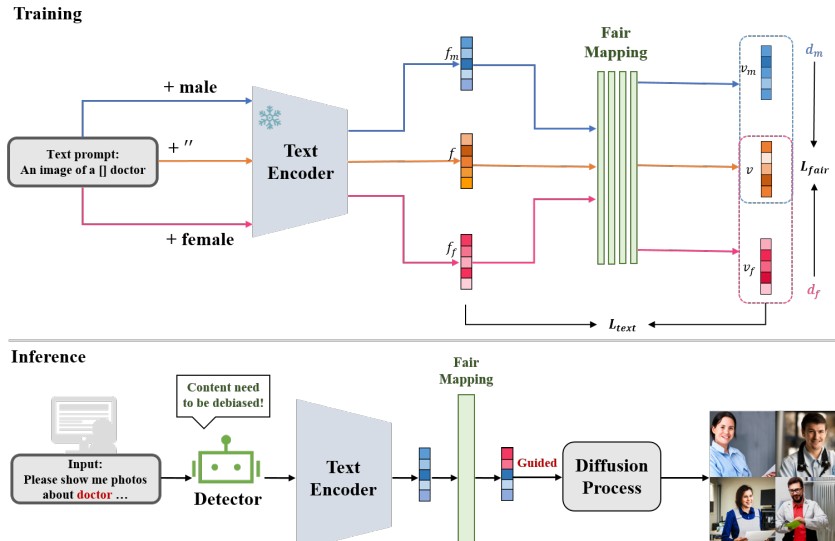

Figure 3: **The figure illustrates our proposed method during both the training and inference stages.** In the training stage, the parameters of the language model are frozen, and we assign weights to loss terms, $\mathcal{L}_{text}$ and $\mathcal{L}_{fair}$ to update Fair Mapping. $d_a$ denotes the distance between $v_a$ and $v$.

To begin with, we introduce the following definitions: $c$ denotes the human-related keyword in the dataset, $A$ represents the sensitive group like *Gender* and *Race*, and $a \in A$ represents a particular sensitive term within the group $A$. Formally, our conditioning is defined as a set of carefully designed prompts $prompt(a, c) = $ "an image of a $a$ $c$." with keywords $c$ extracted from an occupational dataset.

The training and inference procedure of Fair Mapping is elucidated in Figure 3. This framework leverages a linear mapping network architecture, drawing inspiration from StyleGan (Karras et al., 2018) and MixFairFace (Wang et al., 2023), albeit with a unique layer size configuration. While the language model assumes the responsibility of furnishing conditional representations based on the provided text prompt, Fair Mapping operates in tandem by determining suitable offsets within the embedding space. This enables the correction of native language semantic features, ultimately leading to their debiasing and alignment with the balanced embedding space.

## 3.1 DEBIASING TRAINING

In our framework, we construct two distinct types of prompts based on the keyword $c$. The first type constitutes the original input prompt, denoted as $prompt('', c)$, where we explicitly exclude any sensitive attributes (represented as ''). It does not prioritize explicit sensitive attribute information during training or inference. In contrast, the second type, $prompt(a_j, c)$, where $a_j \in A$, incorporates sensitive words. These prompts are designed to quantitatively explore the language relationship between sensitive attributes and keywords. By explicitly introducing sensitive attributes into the prompts, we aim to probe how the model's behavior aligns with different aspects of fairness.

We extract language embeddings $f$ and $f_j$ from $prompt('', c)$ and $prompt(a_j, c)$ using the language model. These embeddings are essential in conventional text-guided diffusion models for generating coherent and contextually relevant text samples. To enhance the fairness aspect of our approach, we introduce a novel architecture called Fair Mapping ($M$). This architecture consists of linear stacking networks that collaborate with the existing model components. We apply the Fair Mapping architecture to transform the original embeddings, producing new representations denoted as:

$$v = M(f), v_j = M(f_j).$$

In this transformation process, our goal is to generate fair, unbiased embeddings with equitable treatment across sensitive attributes. During training, the objectives of $v$ and $v_j$ are two-fold: 1) They should maintain semantic consistency akin to $f$ and $f_j$, serving as contextual information. 2) Importantly, $v$ should equalize the representation of different demographic groups and prevent the encoding of societal biases that may be present in the original embeddings. Therefore, we employ

bias-aware objectives and regularization techniques to guide the model in generating embeddings that are free from sensitive information, promoting fairness in data representation.

**Text Consistence:** Our approach is designed to maintain consistency and semantic coherence between the original embeddings and mapped embeddings. To achieve this crucial objective, we employ a strategy geared towards minimizing the disparity between pre-transformed and post-transformed features in the embedding space. Specifically, we adopt the mean squared error (MSE) as a metric to measure the reconstruction error, drawing inspiration from the reconstruction rule proposed in decoder architecture (Kingma & Welling, 2014). By applying this metric, we compute a semantic consistency loss for each keyword:

$$\mathcal{L}_{text} = \frac{1}{|A| + 1} \left( ||v - f||^2 + \sum_{a_j \in A} ||v_j - f_j||^2 \right) \tag{1}$$

Through the minimization of this loss, we strive to ensure that the mapped embeddings preserve the crucial information and semantic attributes inherent in the original embeddings. This process safeguards the fidelity and integrity of the data throughout the mapping transformation.

**Fair Distance Penalty:** Proximity of embeddings to groups with sensitive attributes can inadvertently encode demographic-related information. For example, if the word "doctor" is closer to males and farther from females in the language model, it may inherently convey gender bias (Chen et al., 2020). To mitigate this issue, we employ a debiasing method that entails the adjustment of associations between sensitive attributes and words during the training process using mapping offsets. The primary objective is to diminish the expression of these associations within word vectors, thereby reducing the potential for biased representations. Through the application of mapping offsets, we seek to counteract the influence of sensitive attributes on word embeddings, fostering a more neutral and unbiased representation.

To equalize the representations of attributes, the objective is to ensure that these adjusted embeddings have similar distances from the native embeddings, thereby reducing the associations related to sensitive attributes in semantic space. The visualization in Figure 3 illustrates this process. In the case where the size of the sensitive group $A$ is 2, we can minimize the difference in distance between the native embeddings, expressed as $|d(v, v_1) - d(v, v_2)|$. Here, $d(\cdot, \cdot)$ represents the Euclidean distance (Dokmanic et al., 2015) between embeddings. However, to address the computational complexity when dealing with a large attribute set $A$ containing multiple sensitive characteristics, we can optimize representation bias by focusing on reducing the variance in the distance between the embeddings. Instead of calculating the difference in distance for each pair of sensitive attribute embeddings, we can consider minimizing the variance in the distance with respect to the average distance between the native embedding and the sensitive embeddings. The fairness loss term, denoted as $\mathcal{L}_{fair}$, can be formulated as follows:

$$\mathcal{L}_{fair} = \sqrt{\frac{1}{|A|} \sum_{a_j \in A} \left( d(v, v_j) - \overline{d(v, \cdot)} \right)^2} \tag{2}$$

Here, $d(v, v_i)$ represents the Euclidean distance between the native embedding $v$ and the specific sensitive attribute embedding $v_i$. $\overline{d(v, \cdot)}$ refers to the average distance between the native embedding $v$ and all the sensitive attribute embeddings $v_j$. By incorporating this fairness loss term into the training objective, we aim to minimize the variance in the distance between the native embedding and the sensitive attribute embeddings $v_j$. This helps to promote equalization in representation and reduce bias associated with sensitive attributes.

To optimize the overall objective, we combine the Text Consistency Loss, denoted as $\mathcal{L}_{text}$ (from Equation 1), with the estimated bias difference from the Fair Distance Penalty (from Equation 2). This results in the following combined loss function:

$$\mathcal{L} = \mathcal{L}_{text} + \lambda \mathcal{L}_{fair}, \tag{3}$$

where $\lambda$ is a hyperparameter balancing text consistency and fairness. By minimizing this combined loss function, we aim to simultaneously ensure text consistency and reduce bias in the representation of sensitive attributes. To optimize the model, we freeze all parameters except for the additional Fair Mapping network. This allows us to specifically train and fine-tune the Fair Mapping network to

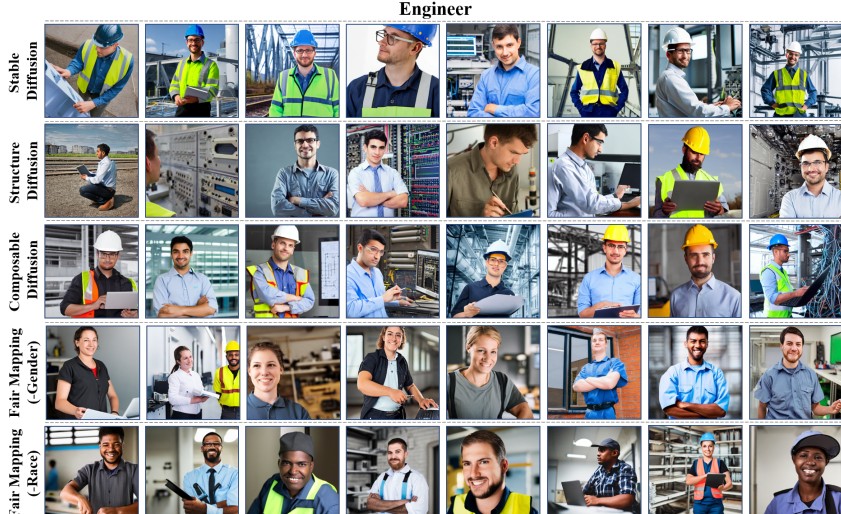

Figure 4: Comparison with different text-to-image methods.

promote fairness and mitigate biases while keeping the rest of the model parameters fixed. By incorporating both the Text Consistency Loss and the Fair Distance Penalty, we aim to achieve a balance between coherence, relevance, and fairness in the generated text samples.

## 3.2 INFERENCE

During inference, before generating text, our approach includes a detector that checks if the input prompt contains predefined sensitive keywords. This detector helps identify if the prompt is related to sensitive attributes. If sensitive keywords like occupation and emotion are detected, we apply the Fair Mapping mechanism to address potential bias in the input prompt. By integrating the detector and Fair Mapping during inference, our approach aims to reduce the generation of biased or sensitive text. This ensures that the model's outputs maintain fairness by avoiding the promotion or reinforcement of biased information in the generated text. See Section C in Appendix for details.

## 4 EXPERIMENTS

Given the limited research and open-source code addressing fairness in diffusion models, there is a scarcity of comparisons with state-of-the-art approaches in this domain. Our evaluations primarily focus on comparing our approach with the prevalent baseline diffusion model, Stable Diffusion (Rombach et al., 2022). We also extend our comparisons to include two text-guided diffusion models: Structured Diffusion (Feng et al., 2023a) and Composable Diffusion (Tang et al., 2023). In this section, we report the performance of our models in three aspects: 1) Our fair mapping method outperforms baselines in fairness evaluation. 2) Fair mapping ensures images prioritize people, showcasing human-related descriptions effectively. 3) Our method matches human preferences in image quality and text alignment of the state-of-the-art text-to-image diffusion method.

## 4.1 EXPERIMENTAL SETUP

**Datasets:** We select a total of 150 occupations and 20 emotions as target keywords for fair generation following (Ning et al., 2023). For sensitive groups, we choose gender groups (male and female) and racial groups (Black, Asian, White, Indian) provided from (Kärkkäinen & Joo, 2021). We provide a comprehensive list of keywords in Appendix D.1.

**Implementation Details:** In our experiments, we use Stable Diffusion (Rombach et al., 2022) trained on the LAION-5B (Schuhmann et al., 2022) dataset and implement 50 DDIM denoising steps. Specifically, we utilize the pre-trained stable diffusion model (SD-1.5). All of our training experiments are conducted on an Nvidia A100 GPU. We maintain a uniform learning rate of 1e-2 and keep the number of training epochs consistent at 500. For each specific occupation and emotion, we set $\lambda$ to 0.1. We set the number of layers to eight for linear mapping structure in Fair Mapping.

**Evaluation Metrics:** We systematically evaluate each method based on two criteria: fairness of text embedding and generative results and quantitative evaluation of generative images using human-related descriptions in the diffusion model. We assess language fairness by incorporating semantic

similarity calculation (Chen et al., 2020; Mikolov et al., 2013) between keywords and sensitive groups. See Appendix D.2.1 for details. Furthermore, we propose a novel evaluation metric rooted in Individual Fairness to robustly assess fairness in generative results of diffusion models across diverse groups. This metric captures variations in generated outcomes among demographic factors (Hardt et al., 2016), such as gender and race, and quantifies fairness by evaluating equilibrium. Our study adopts a highly specific and constrained definition of fairness in the evaluation process. In detail, given any keyword $c_k$ in the dataset $\mathcal{D}$ with its possible sensitive attributes $\mathcal{S}_k$, the diffusion model is absolutely fair if it satisfies

$$P(A = s_i | c = c_k) = P(A = s_j | c = c_k), \text{for all } s_i, s_j \in \mathcal{S}_k, \tag{4}$$

where $A$ represents the sensitive attribute random variable, $c$ is conditional prompt used to guide generative images, $P(A = s_i | c = c_k)$ represents the probability of the sensitive attribute $A$ of generative images expressing $s_i$ given the specific conditional prompt $c = c_k$. Thus, based on equation 4, for a keyword $c_k$, our fair evaluation metric on the diffusion bias is designed as follows:

$$\text{FairScore}(c_k) = \sqrt{\frac{1}{|\mathcal{S}_k|} \sum_{s_i \in \mathcal{S}_k} \left( P\left(A = s_i \mid c = c_k\right) - \frac{1}{|\mathcal{S}_k|} \sum_{s_j \in \mathcal{S}_k} P\left(A = s_j \mid c = c_k\right) \right)^2}. \tag{5}$$

Thus, for a dataset $D$ that contains keywords, our fair evaluation metric on the diffusion bias is $\frac{1}{|\mathcal{D}|} \sum_{c_k \in \mathcal{D}} \text{FairScore}(c_k)$. A smaller value of the metric indicates that the method is more fair.

We present a quantitative evaluation of the image quality generated by the models after applying Fair Mapping to different sensitive groups. To measure the alignment between generated images and human-related content, we calculate the CLIP-Score Hessel et al. (2021), which measures the distance between input textual features and generated image features. Due to the limitation (Otani et al., 2023) of capturing the specific nuances of human-related textual generation, we introduce the Human-CLIP metric, which focuses specifically on evaluating the CLIP-score related to human appearance for effective assessment of the alignment between the generated images and the human-related content. For diversity, intra-class average distance (Le & Odobez, 2018) is aimed to evaluate the diversity of generative results. More details are included in Appendix D.2.2 and D.2.3

## 4.2 FAIRNESS EVALUATION

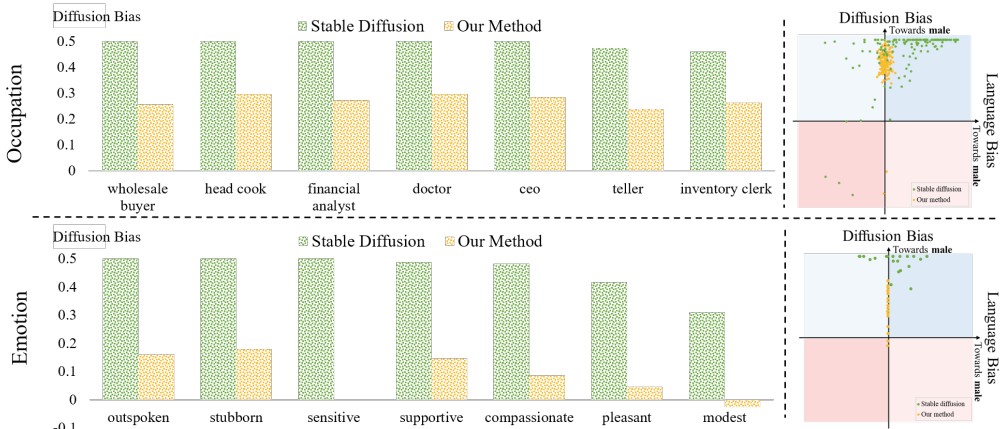

Figure 5: **Left: The comparison results between our method and Stable Diffusion for some keyword examples.** It showcases the disparity between our optimized results and the original results. **Right: The comparison results between our method and Stable Diffusion regarding language bias and diffusion bias on two datasets.** Each point in the figure represents a keyword.

Notably, existing methods exhibit a pronounced lack of female representations in the generated engineer images in Figure 4. This disparity can be attributed to the societal gender stereotypes associating engineers with males. Consequently, the cognitive bias inherent in language prompts the model to generate male engineers more frequently while underrepresenting female engineers. In contrast, our method, when compared to existing text-guided diffusion approaches, demonstrates

an enhanced capacity for generating diverse sensitive attributes while upholding the stability of the generated results. Our generated outcomes exhibit a more substantial representation of marginalized groups, including women and dark-skinned individuals. This highlights the potential of our approach in advancing fairness, inclusivity, and representation in human-related description generation tasks.

**Language bias:** Our investigation affirms the mitigation of bias in the text embedding space. Figure 5 provides a comparative analysis of gender bias in text prompts between our method and Stable Diffusion. Within the occupation dataset, our method notably exhibits significantly reduced language bias compared to Stable Diffusion, indicating a reduced bias toward specific keywords. Likewise, our method surpasses Stable Diffusion in reducing language bias within the emotion dataset.

**Fair Face Generation:** Figure 5 illustrates the comparative results of bias detection on selected keywords between our method and the stable diffusion method. Positive values indicate a male gender bias in the generated image results. While these may still be influenced by factors such as the diffusion process or decoder-related biases, efforts have been made to mitigate these effects. We conduct a comprehensive comparison of our experiment with baselines in sensitive group *Gender* and *Race*, evaluating their performance in terms of our proposed diffusion fairness metrics. Table 1 presents a summary of the results in *Gender* and *Race*, respectively. We observe that our method consistently outperforms other methods and achieves a higher fairness performance, indicating a more equitable and unbiased outcome compared to the alternative approaches tested. These results highlight the effectiveness of our method in promoting fairness and mitigating bias.

Table 1: Fair evaluation results of sensitive group gender and race. O denotes the Occupation dataset and E denotes the Emotion dataset.

| Dataset | Models | Diffusion Bias (O) | Diffusion Bias (E) |
|---|---|---|---|
| Gender | Stable Diffusion | 0.4466 | 0.4652 |
| | Structure Diffusion | 0.4141 | 0.4100 |
| | Composable Diffusion | 0.4027 | 0.4203 |
| | Fair Mapping (Ours) | **0.3625** | **0.2113** |
| Race | Stable Diffusion | 0.2599 | 0.1893 |
| | Structure Diffusion | 0.2368 | 0.1824 |
| | Composable Diffusion | 0.2344 | 0.1489 |
| | Fair Mapping (Ours) | **0.2231** | **0.1178** |

### 4.3 QUANTITATIVE ANALYSIS

**Alignment:** In Table 3, we compare our method with all baselines for images generated with human-related descriptions. While our method shows a slight decline in the CLIP-Score compared to the baselines, it performs superior performance in terms of the Human-CLIP metric. Therefore, our method's effectiveness in the Human-CLIP metric highlights its success in capturing human-related characteristics in generated images despite a slightly lower overall CLIP-Score.

**Diversity:** Diversity refers to the range of image environments and the degree of variation in the generated images' environmental attributes. Our method achieves favorable improvement in terms of metrics that evaluate the diversity of generated results compared to Stable Diffusion, which means it excels in generating diverse and varied results. More visual examples in F.

**Ablation Study:** Table 2 shows the results of an ablation study examining the influence of different factors in the loss terms, $\mathcal{L}_{text}$ and $\mathcal{L}_{fair}$, on model performance. We implement our experiments in Group *Gender* and dataset *occupation*. The table reveals that $\mathcal{L}_{text}$ can function independently, as indicated by individual rows representing the method's performance when only one of these criteria is considered. However, it is evident that $\mathcal{L}_{fair}$ alone is not

Table 2: An ablation study on $\mathcal{L}_{fair}$ and $\mathcal{L}_{text}$ in the loss function. O denotes the Occupation dataset and E denotes the Emotion dataset.

| $\mathcal{L}_{text}$ | $\mathcal{L}_{fair}$ | Diffusion Bias(O) | Diffusion Bias(E) |
|---|---|---|---|
| - | - | 0.4466 | 0.4622 |
| - | ✓ | - | - |
| ✓ | - | 0.4030 | 0.3862 |
| ✓ | ✓ | **0.3624** | **0.2113** |

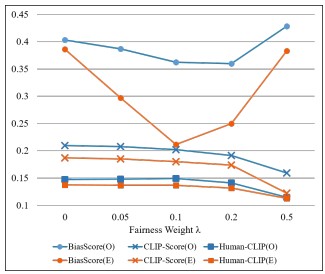

Figure 6: **The influence of $\lambda$ values**. (O) means Occupation dataset and (E) means Emotion dataset.

Table 3: Evaluation results of image alignment and diversity.

| Models | Occupation | | | Emotion | | |
|---|---|---|---|---|---|---|
| | CLIP-Score | Human-CLIP | Diversity | CLIP-Score | Human-CLIP | Diversity |
| Stable Diffusion | 0.2320 | 0.1339 | 13.61 | **0.2026** | **0.1399** | 1.39 |
| Structure Diffusion | **0.2339** | 0.1284 | 13.31 | 0.1930 | 0.1103 | 1.50 |
| Composable Diffusion | 0.2299 | 0.1367 | 12.82 | 0.1901 | 0.1155 | **1.52** |
| Fair Mapping-Gender | 0.2021 | 0.1494 | 14.07 | 0.1809 | 0.1366 | 1.47 |
| Fair Mapping-Race | 0.2197 | **0.1522** | **14.14** | 0.1848 | 0.1324 | 1.41 |

effective and requires the presence of $\mathcal{L}_{text}$ to establish an effective semantic space. The combination of generating diverse sensitive attributes ($\mathcal{L}_{text}$) and maintaining fairness in representation ($\mathcal{L}_{fair}$) achieves the lowest diffusion bias, indicating superior performance in terms of fairness.

Figure 6 shows how $\lambda$ influences the quality and fairness. $\lambda$ parameter to adjust the weight of fairness loss can balance quality and fairness in image generation tasks. Smaller $\lambda$ values provide more emphasis on alignment, producing visuals that are more similar to text descriptions but may miss fairness. Larger $\lambda$ values put fairness first and respond to the needs of vulnerable populations, but they degrade image quality. When $\lambda$=0.5, images suffer severe distortion, containing only limited semantic information, leading to a decline in fairness as well.

## 4.4 HUMAN PREFERENCE

We conduct a human study about the fidelity and alignment of our method (details are included in Appendix E.3). Table 4 shows the results of our statistic of human preference scores. For fidelity, the human preference scores reveal that our method consistently outperforms the other generative images both for occupation and emotion description. Our method prioritizes fidelity in generating images when given a simple human-related description. Meanwhile, our method has a slightly lower alignment between the generated images and the input text features in comparison to other text-to-image approaches in descriptions of emotion. Our method introduces a trade-off between prioritizing fairness and maintaining the alignment of facial expressions and textual descriptions. Therefore, according to our user survey, some participants expressed dissatisfaction with our method' performance in achieving consistency between facial expressions and textual descriptions. Future research endeavors may focus on enhancing the consistency of the text prompts while balancing the bias.

Table 4: Evaluation Results in Human Preference. The higher the score, the more it aligns with human preferences. Please refer to the Appendix E.3 for the scoring criteria.

| Models | Occupation | | Emotion | |
|---|---|---|---|---|
| | Fidelity | Alignment | Fidelity | Alignment |
| Stable Diffusion | 2.7558 | **3.6760** | 2.7230 | 3.4929 |
| Structure Diffusion | 2.5399 | 2.9953 | 3.3427 | 3.3615 |
| Composable Diffusion | 2.6667 | 3.0375 | 1.9718 | **3.6854** |
| Fair Mapping-Gender | 3.0140 | 3.0760 | 3.4883 | 3.2431 |
| Fair Mapping-Race | 3.0798 | 3.3661 | 3.0140 | 3.3475 |
| Real Image | **3.4694** | - | **3.5576** | - |

## 5 DISCUSSION AND CONCLUSION

In this study, we propose that inherent biases in language contribute to the observed bias in text-guided diffusion models, underscoring the potential of language models to introduce and amplify biases during the process of text generation and text-guided diffusion. To combat language biases, we develop a method that effectively mitigates bias within the text space with minimal additional training parameters. Furthermore, we introduce new fairness evaluation metrics, demonstrating substantial improvements when compared to other text-guided diffusion models. Despite our efforts to address language biases, the complete elimination of generation biases remains challenging, indicating the influence of factors beyond language models, with intertwined information from diverse modalities within the diffusion model. Looking ahead, our future research endeavors will delve into the complex issue of bias entanglement across different modalities, aiming for a more comprehensive understanding and mitigation of biases in diffusion models. This research has the potential to greatly enhance the fairness of generative models in various domains, including social media and recommender systems, where diffusion models have a significant societal impact.

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

# A   PRELIMINARIES

Diffusion models are generative models that generate images by iteratively applying denoising steps, starting from a random noise map. The process involves gradually refining the initial noise map to produce high-quality images. In a diffusion model, the generation process is often represented by a diffusion equation or a sequence of denoising steps. The initial noise map, denoted as $z$, serves as the input. The model then applies a series of denoising steps, progressively reducing the noise and improving the image quality. A predefined number of denoising steps determines the degree of noise at each step and a timestep-dependent noise prediction network $\epsilon_\theta$ is trained to predict the noise added to a given input $z$.

Although earlier models, such as Denoising Diffusion Probabilistic Models (DDPM) (Ho et al., 2020), are computationally expensive, the non-Markovian diffusion method, Denoising Diffusion Implicit Models (DDIM) (Song et al., 2021), has improved the inference speed by drastically reducing the number of denoising steps. In DDIM, the noise prediction network $\epsilon_\theta$ is utilized to estimate the noise added at each denoising step. By reducing the number of denoising steps, DDIM achieves faster inference without compromising the quality of generated images. This improvement in computational efficiency allows for more practical and scalable implementation of diffusion models in various applications.

Text-to-image diffusion models involve the utilization of diffusion models in combination with textual descriptions to generate image samples. The goal of this process is to produce images that correspond to the given conditional textual information, represented as $pt$, thereby establishing an explicit and controllable condition for image generation. According to the sampling process of DDIM,

$$\boldsymbol{x}_{t-1} = \sqrt{\alpha_{t-1}} \underbrace{\left( \frac{\boldsymbol{x}_t - \sqrt{1-\alpha_t}\epsilon_\theta^{(t)}\left(\boldsymbol{x}_t \mid pt\right)}{\sqrt{\alpha_t}} \right)}_{\text{predicted } \boldsymbol{x}_0} + \underbrace{\sqrt{1-\alpha_{t-1}} \cdot \epsilon_\theta^{(t)}\left(\boldsymbol{x}_t \mid pt\right)}_{\text{direction pointing to } \boldsymbol{x}_t}, \tag{6}$$

Where $\boldsymbol{x}_{t-1}$ represents the previous sample at time step $t-1$, $\alpha_{t-1}$ denotes the diffusion parameter at time step $t-1$, $\boldsymbol{x}_t$ is the current sample at time step $t$, $\epsilon_\theta^{(t)}\left(\boldsymbol{x}_t \mid pt\right)$ denotes the noise added to the current sample at time step $t$, which is parameterized by $\theta$ and conditioned on $pt$. $\epsilon_\theta^{(t)}\left(\boldsymbol{x}_t \mid pt\right)$ holds the key to generate images with conditions. Finally, it aims to learn the conditional generation probability density function $p(\boldsymbol{x}_t \mid pt)$.

# B   LANGUAGE BIAS IN TEXT-TO-IMAGE DIFFUSION MODELS

We conduct the following experiment in the occupation keyword set in Appendix D.1: 1) First, we calculate language bias on every occupation in the keyword dataset over sensitive attribute gender. 2) Then, for each occupation $c$, we use the following prompt format: "an image of a $c$" for guiding stable diffusion model to generate 100 images and measure the diffusion bias. Figure 2 shows the experimental results.

In the left region of Figure 2, we conduct a language bias analysis on specific occupations and find that the language used to describe these occupations exhibits gender bias, aligning with societal stereotypes. As an illustrative instance, our examination of the term *aerospace engineer* reveals a pronounced inclination towards males, which is consistent with the gender bias cognition in the real world towards aerospace engineers.

As illustrated in the left-hand side of Figure 2, we employ a scatter plot representation, where the y-axis corresponds to diffusion bias and the x-axis represents language bias. Notably, the majority of the occupations exhibit a discernible language bias favoring males, as evidenced by the clustering of data points toward positive values on the x-axis. Additionally, there is a pervasive diffusion bias observed across nearly all occupations, except for those specifically associated with women, such as "waitress" and "actress". The majority of data points are concentrated in the region where both language bias and diffusion bias exhibit male bias. This suggests a mutual reinforcement between language bias and diffusion bias. When there is male bias present in language, it may further propagate and influence the results generated by the diffusion model, leading to the formation of diffusion bias. However, there are a few data points that show bias towards males in diffusion

bias while having bias towards females in language bias. This indicates that the causes of diffusion bias may be more complex in these specific cases, potentially involving other factors or mechanisms such as decoder biases, which contribute to this inconsistency. In summary, language bias is one of the direct factors leading to diffusion bias. By addressing and mitigating language bias, it is possible to reduce the impact it has on diffusion bias and promote more equitable and unbiased generative outcomes.

## C    DETAILS OF THE INFERENCE STAGE

In the inference stage, Fair Mapping should keep in robustness to meet requirements of possible de-biasing content. For example, Fair Mapping should be activated whatever the user's prompt could be "I want to show an a $c$ figure", "An image of a $c$" and other formats containing keyword with implicit bias. To ensure reliability across diverse descriptions, an additional detector is introduced with the primary objective of adapting the user's input prompt to a training prompt that exhibits the closest semantic similarity. To achieve this, we calculate the similarity distance between the input prompt and each training prompt of the linear network using a pre-trained text encoder. Subsequently, we identify the training prompt that exhibits the smallest distance. If the calculated distance falls below a pre-defined threshold, we transform the input prompt to match the identified training prompt.

However, an additional issue may arise. The linear network aims to debias the implicit bias associated with the prompt without explicit biased information. It can easily misunderstand explicit biased information in input text, such as "An image of a male doctor", where the use of the linear network becomes unnecessary. Therefore, it becomes imperative for the detector to identify the presence of any sensitive attribute in the transformed training prompt. If the closest training prompt lacks sensitive attributes, passing it through the linear network for debiasing becomes necessary. Conversely, if sensitive attributes are present, skipping the linear network is warranted. The detailed algorithm for the detector is provided in Algorithm 1.

---

**Algorithm 1:** Detector Processing Algorithm

---

**Input:** Input textual prompt $w$, a threshold $e > 0$, a keyword training set $C$ with sensitive
        attribute set $A$, training prompt set $S$ for $C$ and $A$
**Output:** Modified text
**for** *Each prompt $s \in S$* **do**
   | Calculate similarity distance $d_s$ = SimilarityDistance($w, s$);
**end**
$id = \arg\min_{s \in S}(d_s)$;
$tp = S[id]$;
**if** $d < e$ **then**
     **if** $tp$ *does not contain any sensitive attribute word in $A$* **then**
       | Return $w$ to the text encoder and skip the linear network;
     **end**
     **else**
       | Return $tp$ to the text encoder;
     **end**
**end**
Return $w$ to the text encoder and skip the linear network;

---

**Discussions.** We can easily see that there are several strengths of Fair Mapping. Firstly, Fair Mapping is model-agnostic, i.e., as it only introduces a linear mapping network and a detector without modifying the retraining parameters of the diffusion models, we can clearly see that our method can easily be integrated into any text-to-image diffusion models. Secondly, Fair Mapping is lightweight. As a post-processing approach, Fair Mapping only introduces an additional linear map that has to be trained and does not involve any update of the weight parameters in the classical diffusion models. Moreover, as we will mention in the experiments, an eight-layer linear network is sufficient to achieve good performance on both utility and fairness. Thus, the linear network training is quite fast and the additional time cost due to Fair Mapping is extremely small. Finally, our method is quite flexible. Due to the simplicity of our loss for each keyword, we can see our linear network can be replaced by any other prompts, loss of semantic consistency, and the loss of fairness.

# D  EXPERIMENTAL DETAILS

Table 5: Keywords about Occupations

| | | |
|---|---|---|
| Accountant | Facilities Manager | Office Worker |
| Aerospace Engineer | Farmer | Painter |
| Aide | Fast Food Worker | Paralegal |
| Air Conditioning Installer | File Clerk | Payroll Clerk |
| Architect | Financial Advisor | Pharmacist |
| Artist | Financial Analyst | Pharmacy Technician |
| Author | Financial Manager | Photographer |
| Baker | Firefighter | Physical Therapist |
| Bartender | Fitness Instructor | Pilot |
| Bus Driver | Graphic Designer | Plane Mechanic |
| Butcher | Groundskeeper | Plumber |
| Career Counselor | Hairdresser | Police Officer |
| Carpenter | Head Cook | Postal Worker |
| Carpet Installer | Health Technician | Printing Press Operator |
| Cashier | Host | Producer |
| CEO | Hostess | Psychologist |
| Childcare Worker | Industrial Engineer | Public Relations Specialist |
| Civil Engineer | Insurance Agent | Purchasing Agent |
| Claims Appraiser | Interior Designer | Radiologic Technician |
| Cleaner | Interviewer | Real Estate Broker |
| Clergy | Inventory Clerk | Receptionist |
| Clerk | IT Specialist | Repair Worker |
| Coach | Jailer | Roofer |
| Community Manager | Janitor | Sales Manager |
| Compliance Officer | Laboratory Technician | Salesperson |
| Computer Programmer | Language Pathologist | School Bus Driver |
| Computer Support Specialist | Lawyer | Scientist |
| Computer Systems Analyst | Librarian | Security Guard |
| Construction Worker | Logistician | Sheet Metal Worker |
| Cook | Machinery Mechanic | Singer |
| Correctional Officer | Machinist | Social Assistant |
| Courier | Maid | Social Worker |
| Credit Counselor | Manager | Software Developer |
| Customer Service Representative | Manicurist | Stocker |
| Data Entry Keyer | Market Research Analyst | Supervisor |
| Dental Assistant | Marketing Manager | Taxi Driver |
| Dental Hygienist | Massage Therapist | Teacher |
| Dentist | Mechanic | Teaching Assistant |
| Designer | Mechanical Engineer | Teller |
| Detective | Medical Records Specialist | Therapist |
| Director | Mental Health Counselor | Tractor Operator |
| Dishwasher | Metal Worker | Truck Driver |
| Dispatcher | Mover | Tutor |
| Doctor | Musician | Underwriter |
| Drywall Installer | Network Administrator | Veterinarian |
| Electrical Engineer | Nurse | Waiter |
| Electrician | Nursing Assistant | Waitress |
| Engineer | Nutritionist | Welder |
| Event Planner | Occupational Therapist | Wholesale Buyer |
| Executive Assistant | Office Clerk | Writer |

Table 6: Keywords about Emotions

| | | |
|---|---|---|
| ambitious | determined | pleasant |
| assertive | emotional | self-confident |
| committed | gentle | sensitive |
| compassionate | honest | stubborn |
| confident | intellectual | supportive |
| considerate | modest | unreasonable |
| decisive | outspoken | |

### D.1 KEYWORD DATASET

In our research, we selected keywords for fair image generation based on a thorough investigation detailed in (Ning et al., 2023). These chosen keywords cover a variety of job roles (refer to Table 5) and emotional states (see Table 6). Our experiments involve a total of 150 different occupations and 20 emotional states, ensuring a diverse and comprehensive range for a thorough examination of our proposed approach.

### D.2 MORE DETAILS ON EVALUATION METRICS

#### D.2.1 LANGUAGE BIAS

We assess language bias by incorporating semantic similarity calculation (Chen et al., 2020; Mikolov et al., 2013) between keywords and sensitive attributes. Specifically, we use Euclidean distance to evaluate the distance between native keyword terms and specific sensitive attribute word terms. The closer distance indicates a potential bias in the language representation towards a specific sensitive word. We define our language bias evaluation criteria towards attribute $a_i$ for keyword $c_k$ and our input prompts as $LBias_{a_i}(c_k)$:

$$LBias_{a_i}(c_k) = -\|f_j - f\|_2 + \frac{1}{|A|} \sum_{a_j \in A} \|f_j - f\|_2, \tag{7}$$

where $\|f_j - f\|_2$ represents the Euclidean distance between the prompt generated with the sensitive term $a_i$ and the keyword $c_k$, compared to the prompt generated with no sensitive term.

#### D.2.2 HUMAN-CLIP

The CLIP (Contrastive Language-Image Pretraining) Score serves as a prominent evaluation metric utilized for the assessment and comparison of semantic similarity between images and text in the context of generative models. This metric entails the utilization of pre-trained CLIP models, where images and text are inputted, and subsequent semantic relatedness is measured based on the similarity scores generated by the model. Higher scores indicate a greater degree of semantic relevance between the image and text, while lower scores suggest diminished semantic coherence. While CLIP demonstrates favorable performance across various tasks and domains, it is not exempt from limitations and challenges. Particularly in the case of text pertaining to human-related decriptions, the intricate and ambiguous nature of language poses difficulties for CLIP in achieving comprehensive understanding. Instances may arise where CLIP produces high scores for captions or input questions that exhibit inconsistency with the textual content, as depicted in the provided Figure 7.

Consequently, when confronted with human-authored descriptions, the stable diffusion may encounter constraints in its capacity to effectively generate images featuring human faces. We assess the efficacy about human face generating of diffusion models, as presented in Table 7. In terms of occupations, the three different diffusion models, namely Stable Diffusion, Structure Diffusion, and Composable Diffusion, achieved frequencies of 0.5914, 0.5641, and 0.6101, respectively, in generating facial images related to occupations. On the other hand, for emotions, the frequencies achieved by the Stable Diffusion, Structure Diffusion, and Composable Diffusion models were 0.6904, 0.5850, and 0.6274, respectively. Our approach demonstrates significant proficiency in generating realistic human facial features within generative images, exhibiting a notable improvement of over 30% in performance when compared to alternative methods.

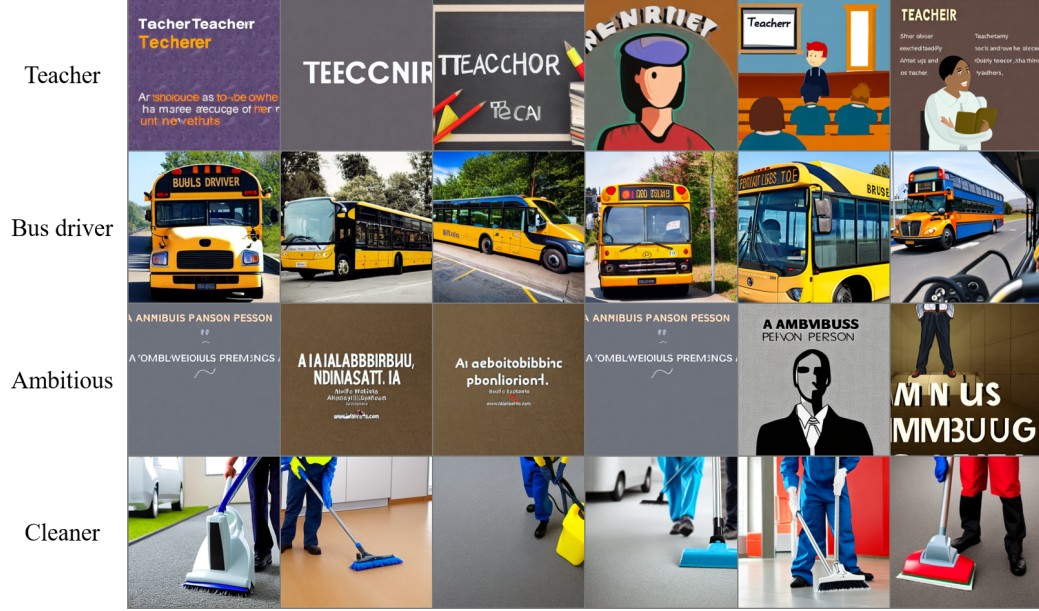

Figure 7: Misalignment between generated images and human-related descriptions in Stable Diffusion.

We propose the Human-CLIP metric as a remedy to the problem of not being able to capture all the subtleties in human-related textual generation based on the viewpoint presented in the literature (Otani et al., 2023). The Human-CLIP measure aims to accurately analyze the alignment between generated images and content that is relevant to humans by concentrating on analyzing the CLIP-Score related to human appearance, offering a way to gauge the effectiveness of text-to-image production and its applicability to humans by using the CLIP model's scores. It gets over the drawbacks of earlier metrics and offers academics and practitioners a new tool for evaluating and comparing the quantitative connection between generated images and human appearance.

Table 7: Evaluation Results in Image Effectiveness for human frequency.

| Models | Occupation | Emotion |
|---|---|---|
| Stable Diffusion | 0.5914 | 0.6904 |
| Structure Diffusion | 0.5641 | 0.5850 |
| Composable Diffusion | 0.6101 | 0.6274 |
| Fair Mapping-Gender | 0.9229 | **0.8950** |
| Fair Mapping-Race | **0.9318** | 0.7237 |

Specifically, we selectively retain the CLIP-Score solely for generative images containing human subjects. For images lacking human descriptions, we assign a CLIP-Score value of 0:

$$\text{Human-CLIP}(img,t) = \begin{cases} CLIP(img,t), \text{if } img \text{ contains human} \\ \qquad\qquad 0, \text{else} \end{cases},$$

Where $CLIP(img,t)$ denotes the CLIP-Score between image $img$ and text $t$. To assess the overall performance of our model, we calculate the Human-CLIP score by averaging the Human-CLIP scores obtained for all generated images. Mathematically, the Human-CLIP score is defined as $\frac{1}{|I|}\sum_{img,t\in I}$ Human-CLIP$(img,t)$, where $I$ is the set of image-text pairs $(img,t)$ in the generated image set. This calculation provides an aggregated measure of the model's ability to generate images that align with human perception and understanding, as evaluated through the CLIP model.

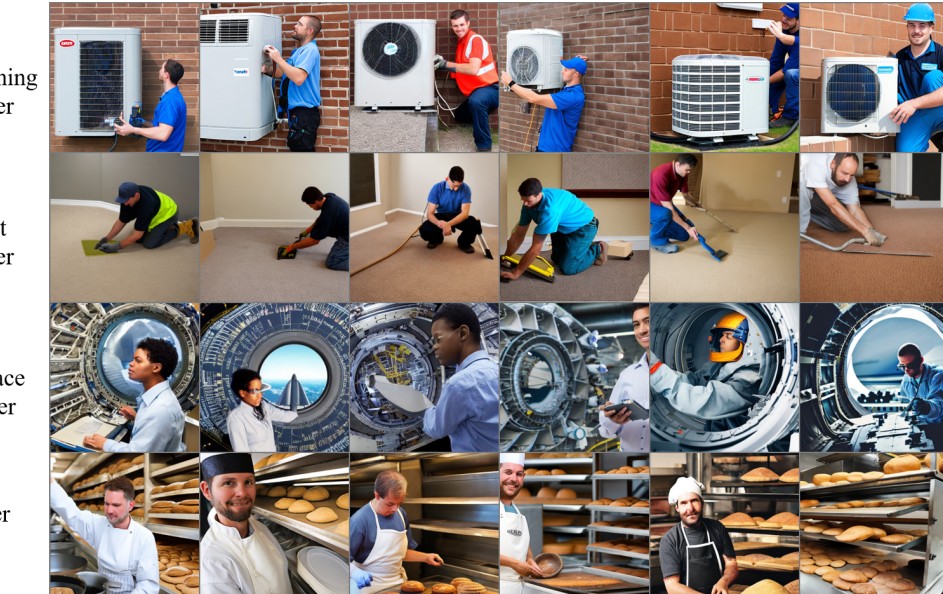

Figure 8: characteristics of repetitive scenes in Stable Diffusion.

### D.2.3 DIVERSITY

Stable Diffusion results show similarities to Mode Collapse in GANs, generating repetitive scenes with consistent backgrounds. As shown in Figure 8, for example, when generating images related to "air conditioning installer", the generated images often depict walls and air conditioning units with repetitive perspectives. We employ the intra-class average distance (ICAD) as a measure to assess the diversity of visual environments. To compute this metric, we focus on a particular class or category of generated images, such as "Teacher" or "Pleasant". By employing the squared Euclidean distance as the distance metric, we determine the average distance between all pairs of generated images within the chosen category:

$$ICAD(c) = \frac{1}{|S_c|} \sum_{e_k \in S_c} \|e_k - \frac{1}{|S_c|} \sum_{e_i \in S_c} e_i\|_2,$$

where $e_k$ and $e_i$ represents an individual generated image within category $c$, $S_c$ denote an image set generating controlled by $C$.

Subsequently, we calculate the average value $\frac{1}{|D|} \sum_{c \in D} ICAD(c)$ across all keywords, where $D$ is a dataset that contains keywords. The average distance calculated within a keyword serves as a measure of dissimilarity or variability among the images belonging to that category. A smaller average distance indicates a higher degree of similarity or compactness among the images, suggesting a lower level of diversity within the visual environments they represent. Conversely, a larger average distance signifies greater variation among the images, indicating a broader range of visual environments captured by the generated images.

## E  MORE EXPERIMENTAL RESULTS

### E.1  TIME CONSUMPTION

As previously highlighted, our method represents a lightweight framework meticulously designed to underscore computational efficiency. In this section, we provide an empirical illustration of the augmented time consumption associated with our approach when juxtaposed with the vanilla Stable Diffusion Model.

Specifically, during the training phase on a single Nvidia V100 device, our methodology exhibits remarkable efficiency by completing the entire process in a mere 50 minutes, encompassing 150

occupations in the sensitive attribute *Gender*. This notably brief training duration underscores the efficacy of our approach in expediting the model learning process while maintaining fidelity to sensitive attributes.

Table 8 offers a comparative examination of the time required to generate 100 images, contrasting the Fair Mapping and Stable Diffusion. Moreover, in the image generation phase, our method showcases a commendable performance, generating 100 images for a single occupation within 434 seconds. This marks a marginal increase of only 10 seconds when compared with the stable diffusion method. The slight increment in generation time further emphasizes the pragmatic viability of our model, as it continues to deliver expedited results while incorporating robust measures to address and preserve sensitive attributes.

### E.2 COMPARED WITH DEBIASING METHOD

In this section, our goal is to compare our approach with debiasing methods Fair Diffusion(FD)Friedrich et al. (2023). Given the limited research on text-to-image diffusion models, existing studies predominantly focus on post-processing techniques. These involve adjusting model outputs after deployment to alleviate biases arising from human instructions. However, a notable drawback of such techniques is the potential for manual control over the system to output only balanced images through manipulation of human instructions. Note that our decision not to compare fairness stems from the acknowledgment that human instruction methods allow for artificial control over the system to produce exclusively balanced images. This manipulation introduces a subjective bias into the system, rendering comparisons of fairness meaningless. We argue that such controlled interventions may distort the system's natural behavior, making fairness assessments unreliable. Consequently, we prioritize other metrics for a more objective evaluation of our model's performance, emphasizing comparison with alternative debiasing methods based on criteria time consumption, image quality, and output diversity.

Figure 9 illustrates visual comparisons between the keywords 'author' for both fair diffusion and our proposed method. The findings reveal that editing operations exert a substantial negative impact on the quality of image generation, notably affecting details such as teeth, which is common in face manipulation work.

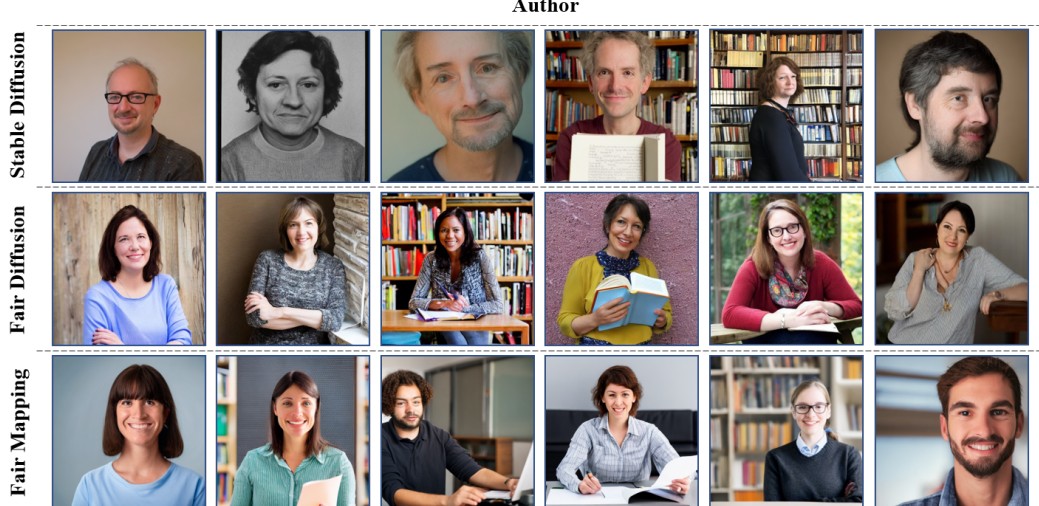

Figure 9: Comparison with different debiasing methods: *Author*

Table 8 provides a comparative analysis of time consumption for generating 100 images between the Fair Mapping and Fair Diffusion models. Additionally, the "Stable Diffusion" baseline is included for reference. All reported time values are in seconds. Our method exhibits superior efficiency, accomplishing the generation of 100 images in a significantly reduced time of 434 seconds. Notably, this represents a substantial time saving of 1029 seconds (approximately 70.3%) compared to the Fair Diffusion model.

Table 8: Evaluation Results in time consumption on generation of 100 images.

| Models | time(seconds) |
|---|---|
| Stable diffusion | 424 |
| Fair Diffusion | 1463 |
| Fair Mapping(our method) | 434 |

Table 9 demonstrates alignment and diversity of the results generated by Fair Diffusion in terms of removing gender and racial biases. In this table, we showcase the generated results by extracting images under equal representation of various sensitive attribute states. Compared with Fair Diffusion, Fair Mapping outperforms Fair Diffusion in terms of alignment with human-related description and diversity. For CLIP-Score, Fair Mapping and Fair Diffusion exhibit relatively similar performance. Under the sensitiva attributes as gender, for Occupation category, Fair Mapping achieves a CLIP-Score of 0.2021, slightly lower than Fair Diffusion. However, Fair Mapping attains a higher Human-CLIP score of 0.1494, surpassing Fair Diffusion's 0.1348. This indicates that Fair Mapping demonstrates better alignment with human-related description. Additionally, in the Emotion category, Fair Mapping achieves a Diversity score of 1.47, exceeding Fair Diffusion's 1.43, highlighting its higher diversity. In the context of addressing race biases, for Occupation category, Fair Mapping surpasses Fair Diffusion in terms of Human-CLIP score, achieving a significantly higher score of 0.1522 compared to Fair Diffusion's score of 0.1292. Furthermore, Fair Mapping exhibits better diversity, scoring 14.14, which is slightly higher than Fair Diffusion's diversity score of 13.79. The significant improvement in both Human-CLIP score and diversity is also observed in the context of addressing race biases for the Emotion category.

Table 9: Evaluation results of image alignment and diversity.

| Models | Occupation | | | Emotion | | |
|---|---|---|---|---|---|---|
| | CLIP-Score | Human-CLIP | Diversity | CLIP-Score | Human-CLIP | Diversity |
| Fair Diffusion-Gender | **0.2274** | 0.1348 | 13.87 | **0.1894** | 0.1298 | 1.43 |
| Fair Diffusion-Race | 0.2239 | 0.1292 | 13.79 | 0.1882 | 0.1266 | 1.39 |
| Fair Mapping-Gender | 0.2021 | 0.1494 | 14.07 | 0.1809 | **0.1366** | **1.47** |
| Fair Mapping-Race | 0.2197 | **0.1522** | **14.14** | 0.1848 | 0.1324 | 1.41 |

### E.3 RESULTS ON HUMAN PREFERENCE

Figure 10 showcases an example of the survey questions and the corresponding results. The evaluation of image authenticity and alignment with textual descriptions was conducted through a survey questionnaire. Participants over 200 from different academic backgrounds are presented with a series of generated images paired with corresponding textual descriptions. They are asked to rate the degree of fidelity in the images and alignment between the images and the provided descriptions from 1 to 5. To evaluate generated images, real images are included as a reference for comparison. For each task, we present users with two sets of 2 images along with the same input conditions for every method in each dataset. To mitigate potential biases stemming from preconceived notions of AI-generated images, we employ a data filtering process to exclude low-scoring samples. Subsequently, we compute the average performance of each method across diverse datasets to derive a comprehensive evaluation. Furthermore, we conducted a short survey to gather feedback on the reasons behind the scoring provided by the evaluators.

Specifically, we demonstrate our score criteria. In this study, we introduce our preference evaluation scale to assess the realism of images, using a scoring range from 1 to 5. The scale is as follows:

- 1 Point: The image is likely AI-generated, displaying obvious artificial characteristics.

- 2 Points: The image may be AI-generated with a realistic style, but discernible discrepancies from real imagery are present.

- 3 Points: The image appears somewhat realistic but may have subtle flaws or unnatural features.

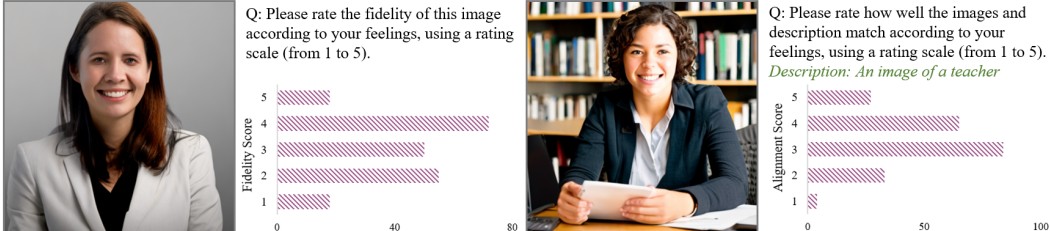

Figure 10: The example for human preference in fidelity and alignment.

- 4 Points: The image is very close to a real photograph, with details, colors, and lighting aligning with the real world.
- 5 Points: The image is indistinguishable from real life, perfectly mirroring real-world standards in every aspect, including details, color, and lighting.

Meanwhile, we employ a scale to evaluate the congruence between images and their corresponding captions, with a rating system ranging from 1 to 5:

- 1 Point: Complete mismatch, the caption does not relate to the image.
- 2 Points: Major discrepancy, the caption largely deviates from the image content.
- 3 Points: Partial difference, there are noticeable mismatches between the caption and the image.
- 4 Points: Minor discrepancy, the caption is almost in line with the image but with slight differences.
- 5 Points: Perfect match, the caption accurately and completely describes the image.

## F  MORE VISUAL RESULTS

In the appendix, visual representations of the generated results are provided to further illustrate the research findings. These displays showcase samples of generated images corresponding to various categories or conditions. Figure 11 and Figure 12 show the images generating from keyword 'CEO' and 'Pleasant' respectively. Upon examination, it is noticeable that there is diversity in the generated images across different sensitive attributes. Specifically, Stable Diffusion tends to generate White male images about keyword 'CEO' and 'Pleasant'. In contrast, our method generating a greater number of female images and images depicting individuals with darker skin tones.

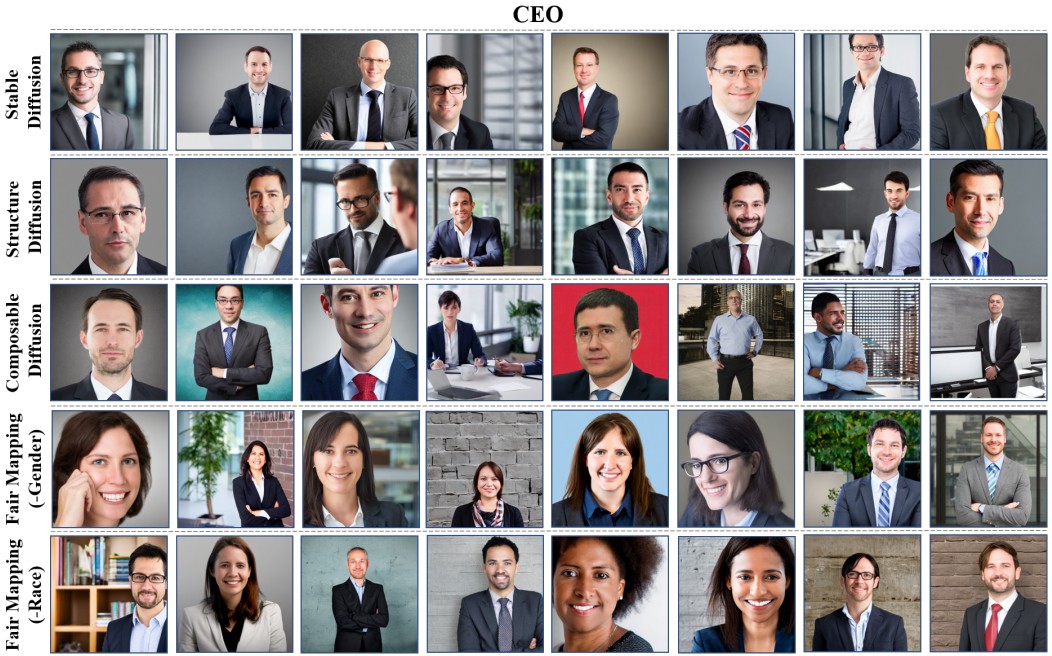

Figure 11: Comparison with different text-to-image methods: *CEO*

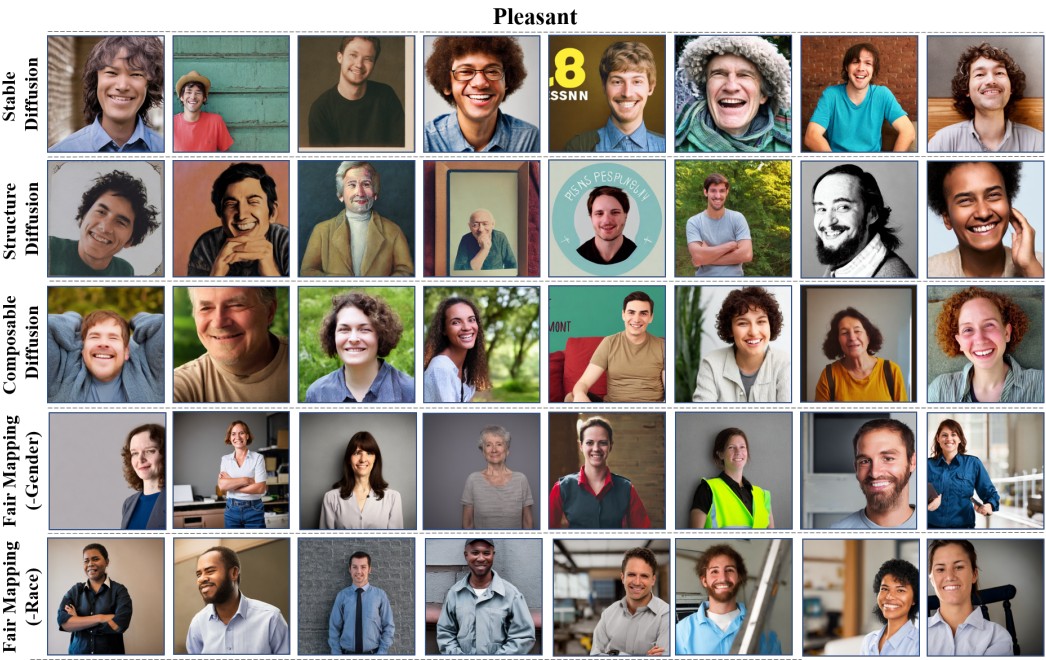

Figure 12: Comparison with different text-to-image methods: *Pleasant*

