# OpenReview forum: "Fair Text-to-Image Diffusion via Fair Mapping"
_ICLR.cc/2024/Conference — Submitted to ICLR 2024_

### Official Review · Reviewer_fsFn · 2023-10-30

**Soundness:** 3 good
**Presentation:** 3 good
**Contribution:** 3 good
**Rating:** 6
**Confidence:** 3

**Summary:**

This paper aims to address the demographic bias in existing text-to-image diffusion models. The paper proposes Fair Mapping, a lightweight module that transforms the input prompts to achieve fair image generation. The training process only involves an additional linear mapping network that projects the language embedding into an unbiased space for generation. The method leads to fairer and more diverse generated images.

**Strengths:**

* This work performs investigations on the bias of text-guided diffusion models and concludes that the main bias related to the generation comes from the language embeddings. This observation can potentially inspire future research in evaluating and relieving biases in text-to-image generation.
* This work proposes a lightweight mapping network that turns the input embeddings that may be biased into unbiased embeddings for generation. This module is simple and easy to understand, yet effective according to the qualitative and quantitative evaluation results.
* This work also proposes a fairness evaluation metric for text-guided human-related image generation, which allows comparing the bias reduction across different de-biasing methods. This could also be useful for future research.

**Weaknesses:**

* This work assumes that the bias can be corrected with an appropriate text embedding, which in turn assumes that the diffusion model is able to generate the specified human characteristics with diverse demographic properties. However, it is possible that the bias in the association is so strong (due to insufficient or biased training data) that the model could not generate the correct image, despite being asked for a specific property. For example, if there are no female plumbers in the training set, the model may not be able to generate the specified image even if "female plumbers" are explicitly asked for in the prompt. Since the method only trains a linear projection head, it is unlikely to be able to generate unbiased images.
* The method relies on a set of predefined sensitive keywords to enable transformation. However, it is hard to make this set of keywords exhaustive, and since this is a matching-based method, the checks may be skipped if some typos are introduced.
* The fair mapping method proposed by the work may lead to loss of details in the prompt, when compared with its baselines, as shown in Sec 4.4 that some facial expressions are not generated according to the text prompts.

**Questions:**

The authors are encouraged to respond and address the weaknesses above.
* If the model itself cannot generate some image characteristics due to insufficient training examples (e.g.,"female plumbers" despite already clearly specified in the text), is the method still applicable?
* How is the set of keywords for activating the linear mapping model defined?

---

> ### Author Response · Authors · 2023-11-19
> **Response to Reviewer fsFn**
>
> We would like to express our gratitude for taking the time to review our work.
>
> > Reply to W1
>
> We admit that if there is a lack of some specific data, then our method may become less efficient. **However, we do not think this is a reason that our paper should be rejected as every post-processing fairness method (our method belongs to this category) will become less efficient even for classical classification tasks if there is a lack of training data.** Thus, all current fairness methods need to assume that the training data is imbalanced or long-tailed and cannot hold the case like "if there are no female plumbers in the training set".
>
>
> > Reply to W2
>
> Yes, you are correct, in our problem we need to pre-define a set of keywords that is used for our training prompts. So our method is tailored for the case where the keyword set is finite. Current research endeavors predominantly revolve around designing limited diverse well-defined benchmarks for evaluation purposes rather than open-world environment.
>
> Regarding typos, there are two cases: (1) if the typo is unintentional, our detector network can somehow correct these typos. In our detector algorithm, we utilize semantic similarity to assess the input. As a result, we are able to effectively handle the issue you mentioned, if semantic errors in the input, such as typos, would not significantly impact the output of the original diffusion model. (2) if the typo is intentional, or it is generated by adversarial attacks, then it is out of the scope of this paper as it is a common issue for all input-to-image diffusion models. It is notable that altering the text embedding is widely used for attackers to manipulate the system in text-to-image diffusion models, which is far beyond introducing mere unfairness in the output. Such attacks pose substantial threats to the integrity and reliability of text-to-image models.
>
> > Reply to W3
>
> Although the fair mapping method may prioritize reducing biases, it can inadvertently sacrifice some fine-grained details or specific prompt-related features, which is also known as the price of fairness. In our work, for image quality, we show that the alignment between generated images and human-related content is very close to that of the vanilla diffusion models. Future work can explore alternative methods or modifications to the fair mapping approach to mitigate the loss of details while still achieving fairness and unbiased image generation.
>
> > Reply to Q1
>
> Please see our response to Weakness 1
>
> > Reply to Q2
>
> In Appendix D.1 of the revised version, we have listed all the keywords we used in the paper. These keywords are carefully chosen to capture the bias inclination associated with bias in human-related descriptions in the context under investigation. It is worth noting that, to the best of our knowledge, at the time of our research, there was no unified benchmark available for this specific task. Therefore, in our experimental setup, the set of keywords used to activate the linear mapping model for debiasing is defined based on statistic of LAION-5B, drawing inspiration from previous work [1].
>
> References
>
> [1] Friedrich F, Schramowski P, Brack M, et al. Fair diffusion: Instructing text-to-image generation models on fairness[J]. arXiv preprint arXiv:2302.10893, 2023.

---

> > ### Author Response · Authors · 2023-11-21
> >
> > Dear Reviewer,
> >
> > Thank you so much for your time and efforts in reviewing our paper. We have addressed your comments in detail and are happy to discuss more if there are any additional concerns. We are looking forward to your feedback and would greatly appreciate you consider raising the scores.
> >
> > Thank you,
> >
> > Authors

---

> > ### Comment · Reviewer_fsFn · 2023-11-21
> >
> > Thanks for the response and updates to the work, which clarifies the work, especially its scope. The reviewer remains positive about this work and still votes for acceptance.

---

### Official Review · Reviewer_4gR2 · 2023-11-01

**Soundness:** 2 fair
**Presentation:** 3 good
**Contribution:** 2 fair
**Rating:** 5
**Confidence:** 2

**Summary:**

This paper introduces the concept of "Fair Mapping" to mitigate bias in text-to-image diffusion models. It addresses the issue of language-induced biases in these models, particularly when generating images based on human-related descriptions. To achieve this goal, in addition to maintain text consistency, the authors also introduce the fairness penalty to encourage unbiased output. By running a set of experiments, the authors show that their approach can significantly reduce the bias in the text embedding space, when compared with other diffusion techniques.

**Strengths:**

The paper introduces a novel method for integrating fairness into conventional diffusion models, offering a valuable means to mitigate language bias in various contexts. The writing in the paper is clear, accompanied by numerous helpful illustrations, making it an enjoyable read.

**Weaknesses:**

One weakness of the paper is that it doesn't clearly elucidate the significance of its results. For instance, it would be beneficial to provide a more in-depth explanation of why we should be concerned with the specific concept of fairness addressed in this paper and to outline potential practical applications of the findings. Moreover, in comparison to other works on fair data generation, it would be helpful to highlight the primary advancements made in this paper and explain their significance. Although I acknowledge that this paper primarily focuses on methodology, I believe that readers would greatly benefit from a clearer understanding of the motivation behind the research and its possible real-world applications. I should note that I am not an expert in this research area, so I may not fully grasp the significance of this work. It's possible that the contributions are evident to experts in this field, but making them more explicit would enhance the paper's accessibility.

**Questions:**

For the abstract and introduction, it would be helpful to provide more motivations and maybe mention the paper's possible applications.

The related work seems a bit unclear to me. I am not familiar with this literature but I feel more detail is needed, especially about previous works on fair data, to understand the contributions of this paper.

---

> ### Author Response · Authors · 2023-11-19
> **Response to Reviewer 4gR2**
>
> Thank you for your feedback and suggestions regarding the motivation of our paper.
>
> > Reply to Q1
>
> In our work, we address the limitations of existing text-to-image diffusion models in generating demographically fair results when given human-related descriptions. For example, the text-to-image diffusion models prefer to generate male images when the input contains a description of ``doctor". We decided to alleviate the implicit bias in language to achieve a fair generation. **To better address your concerns, in the revised version, we have reorganized the paper: (1) We reorganized the introduction part to give a better motivation and significance of our work. Please refer to Page 2 about modified introduction in Section 1. (2) We add one more section on the details of the existence of language bias for text-to-image diffusions. Please see more in Appendix B on Page 15.** Hope this can address your concerns.
>
> > Reply to Q2
>
> Thank you for your feedback, lack of clarity in the related work section. In order to provide a better understanding of the contributions of the paper, I have revised the related work section to include more details about previous works on fair data. I have provided a comprehensive overview of the pre-processing, intra-processing, and post-processing approaches in the related literature. Furthermore, **I have specifically compared our work, which falls under the post-processing category, with the current mainstream post-processing methods for fair data generation.** By highlighting these comparisons, we can better illustrate the distinctiveness and advantages of our approach in achieving fair data generation. This will help the reader, who may not be familiar with this literature, to grasp the context and significance of the paper's contributions.

---

> > ### Author Response · Authors · 2023-11-20
> >
> > Dear Reviewer,
> >
> > Thank you so much for your time and efforts in reviewing our paper. We have addressed your comments in detail and are happy to discuss more if there are any additional concerns. We are looking forward to your feedback and would greatly appreciate you consider raising the scores.
> >
> > Thank you,
> >
> > Authors

---

> ### Author Response · Authors · 2023-11-22
>
> Dear Reviewer 4gR2,
>
> As the author-reviewer discussion period will end soon, we will appreciate it if you could check our response to your review comments. This way, if you have further questions and comments, we can still reply before the author-reviewer discussion period ends. If our response resolves your concerns, we kindly ask you to consider raising the rating of our work. Thank you very much for your time and efforts!

---

### Official Review · Reviewer_ujPt · 2023-11-01

**Soundness:** 3 good
**Presentation:** 3 good
**Contribution:** 3 good
**Rating:** 6
**Confidence:** 3

**Summary:**

This paper introduces a method that addresses diversity limitations of text-to-image diffusion models caused by language biases. The proposed approach, Fair Mapping, involves training a mapping network that tries to preserve the original semantics specified in the text while increasing the representation of sensitive groups in the generated images. The mapping network is lightweight as it operates on embeddings from a frozen text encoder. The authors evaluate their method on human-centric generation and demonstrate that Fair Diffusion improves upon biases rooted in occupations and emotions.

**Strengths:**

- The proposed method is simple and easy to understand.
- The mapping network is trained on top of a frozen text encoder, making it widely applicable to other text-conditioned models.
- The experiments demonstrate that Fair Diffusion reduces language biases and improves generation of more diverse people while maintaining the semantics outlined in the input prompt.
- Ablation studies are provided to highlight the significance of both loss terms in the training objective.

**Weaknesses:**

- In Table 1, the delta in improvement of Fair Mapping over the baselines is relatively small for race. It is difficult to understand how much a 0.01 improvement actually looks like in terms of qualitative performance.
- The authors mention that the value of the loss weight hyperparameter can affect the visual quality. Including image quality metrics like FID would be helpful to quantify how much degradation is introduced because of the debiasing network.
- Some details that are necessary for clarity and context in the main paper have been moved to the appendix. For example, Human-CLIP is a newly introduced metric but the details are not discussed at all in the main paper. Additionally, no context for the human preference evaluations is provided (e.g., at the minimum, clarify whether higher or lower is better in Table 4).
- It is unclear why no evaluations are provided in the second row of Table 2 (L_{fair} only).

**Questions:**

- How is the mapping network initialized? In row 3 of Table 2, it is unclear why L_{text} alone would lead to more "fair" results since it is just trying to minimize the projected embedding to the original. Shouldn't the performance be the same as row 1 if the loss is optimized?
- Have the authors experimented with just optimizing d(v, v_j) rather than (d(v, v_j) - \bar{d(v, -)}) in Equation 2?
- For clarity, is there a separate Fair Mapping module for each occupation and emotion, or is there one shared across all?

---

> ### Author Response · Authors · 2023-11-19
> **Response to Reviewer ujPt (part1)**
>
> We would like to extend our heartfelt gratitude to you for your valuable feedback and constructive suggestions.
>
> > Reply to W1
>
> Compared to the baseline model, our method shows a minimum improvement of 10\% and a maximum improvement of 54\%. Due to the broad definition of race and the high efficiency of diffusion models in generating faces based on occupations, the model tends to exhibit results that are more fair in terms of race compared to gender. Therefore, achieving a 10\% improvement is actually a significant amount.
>
> > Reply to W2
>
> The reason why we do not apply FID is that we lack real images to compare and the Inception network is not suitable for our human-related tasks. 1)FID is is a measure that calculates the distance between the feature vectors of a real image and the generated image, widely used for reconstruction images after training all parameters in generative images, which requires real images in training data. However, in our method, we just finetune a few parameters with prompts rather than thousands of real images. 2) FID is a measure of the distance directly pressed by the multivariate normal distribution, but the extracted generative image features, like our human-related tasks, do not necessarily conform to the multivariate normal distribution.
>
> Therefore, we use a traditional metric, CLIP, to evaluate our alignment between text and generated images. To address the problem that CLIP is not suitable for human-related description work, we have designed the Human-CLIP method, which addresses the alignment issues that CLIP is not suitable for in face generation tasks. Although our method has a decrease in CLIP Score, we have demonstrated that our method performs better in face generation tasks and has overcome the limitations of existing evaluation metrics in Human-CLIP.
>
> > Reply to W3
>
> Due to the limited space, some details about metrics about image qualities are discussed in the Appendix. Human-CLIP is specifically designed to evaluate the quality of generated images based on human perception. It leverages the knowledge and capabilities of CLIP. The scale for the score of Human Preference is displayed to provide a reference for interpreting the scores assigned by human evaluators during preference ranking tasks. The specific details of the scale, including its range and corresponding interpretations, are provided in the Appendix section of the research document. We will also give an easy-to-understand statement of these problems in the late revised version paper. **Please see Page 18 about D.2 and Page 22 about E.4 in Appendix. We will move them to the main text in the revised version if additional pages are allowed.**
>
> > Reply to W4
>
> The reason is stated in the section 4.3 about the ablation study and you can refer to it kindly. The overall efficacy of our optimization is compromised in the absence of $L_{text}$, as the total loss cannot adequately capture the semantic content embedded in textual representations. The diffusion models, devoid of $L_{text}$, tend to produce images characterized by the absence of crucial semantic details, resulting in blurriness and distortion. Consequently, assessing metrics for these experiments becomes superfluous under such conditions.

---

> > ### Comment · Reviewer_ujPt · 2023-11-22
> >
> > Thank you for your clarifications of the weaknesses and questions raised in the original review. For the response to W4, I did note the justification in section 4.3 but I felt the metrics could still be provided as a point of reference. Regardless, the original rating will hold.

---

> > > ### Author Response · Authors · 2023-11-23
> > > **Official Comment by Authors**
> > >
> > > Dear Reviewer ujPt,
> > >
> > > For the response to W4, I have further clarification to provide. At this stage, when the image is distorted to the extent that it no longer contains **recognizable objects or basic semantic information**, including gender, it becomes challenging to conduct fairness comparisons. The images **now consisting of patterns or random noise** lack the necessary semantic content for assessing fairness. When the images no longer contain this attribute information, fairness comparisons become impractical.
> > >
> > > If you have any other questions or need further discussion, please feel free to let me know. Hope you have a nice day!

---

> ### Author Response · Authors · 2023-11-19
> **Response to Reviewer ujPt (part2)**
>
> > Reply to Q1
>
> In this paper, We employ the widely used initialization method of Random Initialization. We observe that when optimizing with $L_{text}$ alone, the generated images exhibit a stronger ability to generate face-related images. Therefore, we believe that by minimizing the difference between the projected embeddings and the original embeddings, there is a greater emphasis on maintaining the consistency of inherent attributes and ignoring sensitive attributes, leading to a reduction in biases among generative results. By minimizing the discrepancy between the projected and original embeddings, the $L_{text}$ loss function encourages the model to prioritize the preservation of non-sensitive features from the original images during image generation. This helps to mitigate over-reliance on sensitive attributes and subsequently reduces the occurrence of related biases.
>
> >Reply to Q2
>
> We have indeed experimented with optimizing only $d(v, v_j)$ in Equation 2, without subtracting $\bar{d(v, -)}$. However, the observed outcomes of these experiments were significantly adverse, leading to distorted generated images and a notable loss of semantic coherence. This suggests that incorporating $\bar{d(v, -)}$ in Equation 2 plays a crucial role in achieving more favorable results, such as maintaining image quality and preserving semantic consistency.
>
> >Reply to Q3
>
> In our experiments, all occupations share a Fair Mapping module and all emotions share another Fair Mapping module. Different detectors are defined as different agents to serve the inputs of users. This implies that we employ multiple detectors to identify and classify the inputs provided by users. Each detector specialize in detecting specific attributes or features related to occupations or emotions. These detectors play a crucial role in the overall system, as they contribute to the inputs that are subsequently processed and mapped by the Fair Mapping modules.

---

> > ### Author Response · Authors · 2023-11-21
> >
> > Dear Reviewer,
> >
> > Thank you so much for your time and efforts in reviewing our paper. We have addressed your comments in detail and are happy to discuss more if there are any additional concerns. We are looking forward to your feedback and would greatly appreciate you consider raising the scores.
> >
> > Thank you,
> >
> > Authors

---

### Official Review · Reviewer_R9zr · 2023-11-01

**Soundness:** 3 good
**Presentation:** 3 good
**Contribution:** 2 fair
**Rating:** 5
**Confidence:** 4

**Summary:**

This paper addresses human-related bias in text-to-image diffusion models, and resolves the issue by proposing a novel fair mapping module which outputs a fair text embedding. Such a module can be trained on top of frozen pre-trained text encoder, and inserting the module during sampling successfully mitigates textual bias. Training the fair module involves two loss terms: (i) text consistency loss, which preserves semantic coherence, and (ii) fair distance penalty, which brings output embeddings within different sensitive groups close together. Further, the authors propose a novel evaluation metric, FairScore, which also plans to achieve the conditional independence of the text prompt and sensitive group information.

**Strengths:**

- The paper tackles a timely and practically-relevant problem supported by a fair amount of experiments. Building fair diffusion models is an area with limited prior research, making this work particularly valuable.
- The proposed method is simple yet effective, and pluggable without modifying the pre-trained model.
- Overall, the paper is clearly written and easy to follow.

**Weaknesses:**

- Although the paper covers a good amount of relevant previous studies, the paper lacks baseline experiments. For example, despite [1] focus on fair-guidance while this work focus on pluggable mapping module, the authors can calculate FairScore and compare w.r.t. training time, overhead memory, etc.
- While the unfairness is largely resolved through the proposed mapping module, such a result may not come at a surprise since FairScore and the employed fairness loss term are quite similar.
- The authors note that a detector network is employed to identify predefined sensitive keywords in the input prompts. There is no additional detailed explanation about the detector network.
- This method explicitly needs a labeled dataset to mitigate the demographic bias in diffusion models. However, in real-world scenarios, it may be challenging to identify and address all potential types of bias comprehensively. Further, there are remaining questions regarding whether it is feasible to (i) simultaneously eliminate multiple types of bias or (ii) sequentially address multiple biases without negatively impacting performance. If such challenges cannot be properly addressed, it would incur a significant amount of training time to erase all types of biases, and heavy memory cost to save all mapping modules corresponding to each bias type.

**Questions:**

- How many random seeds are used throughout the experiments?

[1] Friedrich et al., “Instructing Text-to-Image Generation Models on Fairness.” 2023.

---

> ### Author Response · Authors · 2023-11-19
> **Response to Reviewer R9zr (part1)**
>
> We would like to express our sincere gratitude to you for your valuable feedback and insightful comments on our work.
>
> > Reply to W1
>
> We acknowledge the importance of including baseline experiments and performance comparisons. It is important to note that during the time of our study, which was conducted several months ago, there was a scarcity of publicly available open-source code in the field of fair diffusion model generation. This limitation made it challenging for us to conduct direct performance comparisons with other approaches. However, we are pleased to find that this work has been made openly accessible, including the accompanying code recently. In the revised version of our paper, we provided a comprehensive comparison with the mentioned work and provided detailed explanations of the comparison methodology. **We elucidated its limitations in the degree of human intervention for systems and shows priority for our superiority in generation speed as demonstrated.**
>
> | Models                  | Time (seconds) |
> |-------------------------|-----------------|
> | Stable diffusion        | 424             |
> | Fair Diffusion          | 1463            |
> | Fair Mapping (our method)| 434             |
>
> **Meanwhile, we also evaluate the alignment and diversity of Fair Diffusion. The experimental results are copied from Table 22 in Appendix E.2.**
>
>
> |         Models        |            | Occupation |           |            |   Emotion  |           |
> |:---------------------:|:----------:|:----------:|:---------:|:----------:|:----------:|:---------:|
> |                       | CLIP-Score | Human-CLIP | Diversity | CLIP-Score | Human-CLIP | Diversity |
> | Fair Diffusion-Gender |   0.2274   |   0.1348   |   13.87   |   0.1894   |   0.1298   |    1.43   |
> |  Fair Diffusion-Race  |   0.2239   |   0.1292   |   13.79   |   0.1882   |   0.1266   |    1.39   |
>
> Please see Page 21 - 22 for more experiments details.
>
> > Reply to W2
>
> **We kindly cannot agree that FairScore and the fair loss are similar.** Note that for any keyword $c_k$, its fairscore is defined as
>
> $$\text{FairScore}(c_k)= \sqrt{\frac{1}{\mid A \mid} \sum_{a_i\in A } \left( DBias_{a_i}(c_k) \right)^2},$$ where $DBias_{a_i}(c_k)=P\left(s=a_i \mid c=c_k\right)- \frac{1}{\mid A \mid} \sum_{a_j\in A} P\left( s=a_j \mid c=c_k\right).$
> And the fair loss is defined by
>
> $$L_{fair} = \sqrt{\frac{1}{|A|} \sum_{a_j \in A} \left( d(v,v_j) - \overline{d(v,\cdot)} \right)^2},$$
>
> where  $d(v,v_i)$ represents the Euclidean distance between the native embedding $v$ and the specific sensitive attribute embedding $v_i$. $\overline{d(v,\cdot)}$ refers to the average distance between the native embedding $v$ and all the sensitive attribute embeddings $v_j$.
>
> **We can see that FairScore is defined by the conditional probability while fair loss is defined by the Euclidean distance of the embedded vectors after being transformed by our linear network.** There are no evidence or theories to support the equivalence or similarity of these two totally different metrics (conditional probability of generated images v.s. Euclidean distance of the embedded vectors).

---

> ### Author Response · Authors · 2023-11-19
> **Response to Reviewer R9zr (part2)**
>
> > Reply to W3
>
> **We have added detailed related content about the detector in the revised version of our paper. Please refer to the detailed exposition and algorithm demonstration on Page 16 in Section C of Appendix. In this section, we provide a detailed exposition of the operational principles of our detector.** We introduce an additional detector that aims to adapt the user's input prompt by finding the training prompt that is semantically consistent with it. The detector calculates the similarity distance between the input prompt and predefined prompt using a pre-trained encoder and selects the training prompt with the smallest distance. Meanwhile, the detector checks whether the transformed training prompt contains any sensitive attributes. If it doesn't, the prompt is processed through the fair mapping linear network for debiasing.
>
> >Reply to W4
>
> We do not think this point is a weakness of our paper. First, we agree on that "in real-world scenarios, it may be challenging to identify and address all potential types of bias comprehensively." Addressing all potential types of bias now is impossible for the classical classification problem even though there are many studies. However, not addressing all bias issues does not mean these papers have weaknesses and should be rejected.
>
> As there is only one[1] work of fairness of diffusion models, we believe it is valuable to study such a critical problem and in this paper, we provide the first post-processing method.  Due to a lack of studies on this problem, it is obviously we cannot address all biases in a single paper, that is too harsh for us.
>
> About your two questions: (i) "simultaneously eliminate multiple types of bias." As we mentioned, even for classification tasks, there is still no work that can achieve this. However, we also need to mention that, since our method is a post-processing approach, it can be directly integrated into any other (future) fair text-to-image diffusion models to further improve fairness. In the revised version, we add more discussions on the strengths of our method in Appendix C on Page 16. (ii) "sequentially address multiple biases without negatively impacting performance." This is also a harsh question, to the best of our knowledge, whether there is a trade-off between utility and fairness is even an open problem for classification tasks.
>
> >Reply to Q1
>
> In our experiments, there is only one seed for all text-to-image diffusion approaches. Following stable diffusion, We fixed the seeds as 42 in the whole process of our experiments. As a result, the seeds may contribute little influence to our results among different approaches.
>
> References
>
> [1] Friedrich F, Schramowski P, Brack M, et al. Fair diffusion: Instructing text-to-image generation models on fairness[J]. arXiv preprint arXiv:2302.10893, 2023.

---

> > ### Author Response · Authors · 2023-11-20
> >
> > Dear Reviewer,
> >
> > Thank you so much for your time and efforts in reviewing our paper. We have addressed your comments in detail and are happy to discuss more if there are any additional concerns. We are looking forward to your feedback and would greatly appreciate you consider raising the scores.
> >
> > Thank you,
> >
> > Authors

---

> > > ### Comment · Reviewer_R9zr · 2023-11-22
> > >
> > > Thank you for the detailed response, particularly adding the details regarding the detector network. Of note, I believe Appendix C, which is added during this rebuttal phase, should be included in the main section. Unfortunately, I disagree with authors in two following aspects.
> > >
> > > - While I appreciate the authors' inclusion of [1] as a baseline upon my suggestion, I still believe there are more baselines which should taken into consideration. Authors argued that there currently exist only a single baseline [1] but I believe the debiasing methods for diffusion models should be included as well, e.g., refer to [2, 3, 4, 5]. They are not explicitly targeting "group fairness" but proposes a more general method to debias a diffusion model. For instance, in Figures 5 and 6 in [5], the authors show that their method can also improve racial and gender diversity.
> > > - Moreover, the authors argued that addressing multiple types of biases at once or sequentially is an overly demanding as to current fairness studies. I disagree with the authors because existing diffusion debiasing methods [2, 3, 4, 5] have already successfully managed such scenarios. For example, [5] tackles I2P datasets (from [2]) effectively addressing multiple types of biases at once.
> > >
> > > Considering the above concerns, I hold the view that the paper, despite its focus on a timely issue and the introduction of a fair algorithm re. group fairness, is not yet ready for publication. Therefore, I will maintain my current rating.
> > >
> > > To clarify, I am *not* suggesting the authors to include every recent work on debiasing/concept-erasing as baselines. However, the current version seems to lack a thorough exploration of prior research. I would be more than happy to hear the other reviewers' on this matter.
> > >
> > > \
> > > __References__\
> > > [1] Friedrich et al., “Instructing Text-to-Image Generation Models on Fairness.” 2023.\
> > > [2] Schramowski et al., "Safe Latent Diffusion: Mitigating Inappropriate Degeneration in Diffusion Models." 2023.\
> > > [3] Brack et al., "SEGA: Instructing Text-to-Image Models using Semantic Guidance" 2023.\
> > > [4] Gandikota et al., "Erasing Concepts from Diffusion Models" 2023.\
> > > [5] Gandikota et al., "Unified Concept Editing in Diffusion Models" 2023.

---

> > > > ### Author Response · Authors · 2023-11-22
> > > >
> > > > Dear Reviewer R9zr,
> > > >
> > > > Thanks for your response and your reference papers, we will include these papers in the revised version.
> > > >
> > > > We have checked these papers and found that references [2]-[4]  do not contain any experiments on enhancing fairness. So we do not think we need to compare these methods as there is no guidance on how to generate diverse human faces in these papers.
> > > >
> > > > For reference [5], yes it contains the fairness evaluation and is quite close to our paper. However, we should notice that the paper was released at the end of August. However, based on the ICLR Reviewer Guide https://iclr.cc/Conferences/2024/ReviewerGuide : "We consider papers contemporaneous if they are published (available in online proceedings) within the last four months. That means, since our full paper deadline is September 28, if a paper was published (i.e., at a peer-reviewed venue) on or after May 28, 2023, authors are not required to compare their own work to that paper". Thus, we do not think lacking of comparison with [5] is a weakness of our paper.
> > > >
> > > > Thanks.

---

> ### Author Response · Authors · 2023-11-22
>
> Dear Reviewer R9zr,
>
> As the author-reviewer discussion period will end soon, we will appreciate it if you could check our response to your review comments. This way, if you have further questions and comments, we can still reply before the author-reviewer discussion period ends. If our response resolves your concerns, we kindly ask you to consider raising the rating of our work. Thank you very much for your time and efforts!

---

> ### Author Response · Authors · 2023-11-23
> **Response to Reviewer R9zr**
>
> Dear Reviewer R9zr,
>
> Thank you for bringing up these points. In addition to the fact that the work you mentioned is **concurrent and contemporaneous work**, we think they do not align with our topic **fair diffusion models**.
>
> 1. Regarding your mention of the introduction of semantic editing[3-5] methods, we think that a direct comparison is not appropriate. Indeed, semantic editing methods can be integrated into text-to-image approaches through some mechanisms like agents, creating a new system for debiasing, similar to the work in [1]. However, it is beyond our scope. These papers are orthogonal to our mehod, which can be integrated in other future work. Such methods are only considered theoretically combinable should be another new research like[1]. Our goal is to propose an alternative post-processing method that aims to mitigate the generated bias by removing linguistic bias. While methods integrating agents have the potential to address bias, we still believe it is not necessary to compare our approach directly with these editing methods.
>
> 2. Safe diffusion[2] does not explicitly address justice-related issues but rather focuses more on generating inappropriate content. Therefore, it is also beyond the scope of our comparison.
>
> Finally, we kindly hope you to reconsider the research significance of this paper, and we would greatly appreciate it if you could consider raising the scores accordingly.
>
> [1] Friedrich et al., “Instructing Text-to-Image Generation Models on Fairness.” 2023.
>
> [2] Schramowski et al., "Safe Latent Diffusion: Mitigating Inappropriate Degeneration in Diffusion Models." 2023.
>
> [3] Brack et al., "SEGA: Instructing Text-to-Image Models using Semantic Guidance" 2023.
>
> [4] Gandikota et al., "Erasing Concepts from Diffusion Models" 2023.
>
> [5] Gandikota et al., "Unified Concept Editing in Diffusion Models" 2023.

---

### Official Review · Reviewer_emxE · 2023-11-04

**Soundness:** 2 fair
**Presentation:** 3 good
**Contribution:** 2 fair
**Rating:** 5
**Confidence:** 4

**Summary:**

The paper discusses the problem of demographic bias in text-to-image diffusion models, which often produce biased images due to sociocultural biases in language. The authors propose a solution called "Fair Mapping," which is a model-agnostic and efficient method that modifies pre-trained text-to-image models to generate fair images. This is achieved by adding a linear mapping network that updates a small number of parameters, thus reducing computational costs and speeding up the optimization process.

**Strengths:**

1. The paper is well-written and easy to follow.
2. The method is intuitive and reasonable.
3. The experimental results seem promising.

**Weaknesses:**

1. This paper only considers bias within text embeddings and does not extend to biases that may be inherent in the diffusion model itself. This limitation is significant as it suggests that the system could be susceptible to manipulation if the text embedding model is altered. A more holistic approach that also scrutinizes and corrects for biases within the diffusion model could potentially offer a more robust and less vulnerable solution.
2. The experiments are limited to biases related to gender and race, omitting other prevalent societal biases such as ageism, socioeconomic status and religious discrimination.

**Questions:**

Refer to the weeknesses mentioned above.

---

> ### Author Response · Authors · 2023-11-19
> **Response to Reviewer emxE**
>
> Thank you for raising an important point regarding the limitations of our experiments.
>
> > Reply to W1
>
> We kindly disagree that only considering bias within text embeddings is a limitation. Comprehensively improving fairness for input-to-image diffusion models is a complicated problem as the bias arises from various resources. If not addressing all potential types of bias in the diffusion model should be rejected, then all previous papers on fairness should be rejected.
>
> Currently, there is only one paper[1] studying this topic which is based on post-processing method. Moreover, mitigating inherent bias by fine-tuning all parameters in the diffusion model would necessitate significant time and computational resources. Our method just employs lightweight network structures that incur minimal additional computational cost when updating the parameters. By examining biases at this level, our intention is to illuminate a critical factor influencing the overall output of the system. Due to its simplicity, our method can be easily integrated into any fair input-to-image diffusion model to further improve fairness. **In the revised version, we have added more discussions on the strengths of our method.** Please refer to Page 16 for an in-depth discussion on our strengths 1) Model-Agnostic Integration: Fair Mapping seamlessly integrates with diverse text-to-image diffusion models without requiring modifications to their retraining parameters. 2) Lightweight Implementation: As a post-processing approach, Fair Mapping introduces a trainable linear map, ensuring computational efficiency and minimal additional time costs. 3) Flexibility: Fair Mapping's simple loss function for each keyword allows for easy substitution with alternative prompts or fairness metrics, enhancing adaptability across different scenarios.
>
> For potential attacks, **we do not think it is an issue that is specified to our method as it is a common issue for all input-to-image diffusion models**. It is notable that altering the text embedding is widely used for attackers to manipulate the system in text-to-image diffusion models, which is far beyond introducing mere unfairness in the output. Such attacks pose substantial threats to the integrity and reliability of text-to-image models. However, this issue of increased vulnerability to attacks is commonly observed in text-to-image diffusion models. It is important to note that the limitation highlighted is inherent in all attacking scenarios of text-to-image models, and our proposed mapping network ensuring that the integrity of the end-to-end system's structure is preserved, does not contribute to the amplification of this risk.
>
> > Reply to W2
>
> We cannot agree with the reviewer's comments. As we mentioned, currently there is only one paper[1] on fairness diffusion models previously. There is a lack of unified benchmarks for evaluating the performance of approaches in this domain. In our research, we focus on investigating gender and race biases due to their well-documented presence and significant impact, as evidenced by numerous studies [1-2]. Additionally, these biases are readily observable in images, making them suitable as typical attributes for our research focus. It is worth noting that our method can be extended to address other sensitive attributes based on user preferences. The flexibility of our approach allows for its deployment and adaptation to various attribute domains, providing a versatile framework for bias mitigation in image analysis.
>
> References
>
> [1] Friedrich F, Schramowski P, Brack M, et al. Fair diffusion: Instructing text-to-image generation models on fairness[J]. arXiv preprint arXiv:2302.10893, 2023.
>
> [2] Wang F E, Wang C Y, Sun M, et al. Mixfairface: Towards ultimate fairness via mixfair adapter in face recognition[C]//Proceedings of the AAAI Conference on Artificial Intelligence. 2023, 37(12): 14531-14538.

---

> > ### Author Response · Authors · 2023-11-20
> >
> > Dear Reviewer emxE,
> >
> > Thank you so much for your time and efforts in reviewing our paper. We have addressed your comments in detail and are happy to discuss more if there are any additional concerns. We are looking forward to your feedback and would greatly appreciate you consider raising the scores.
> >
> > Thank you,
> >
> > Authors

---

> > > ### Author Response · Authors · 2023-11-22
> > >
> > > Dear Reviewer emxE,
> > >
> > > As the author-reviewer discussion period will end soon, we will appreciate it if you could check our response to your review comments. This way, if you have further questions and comments, we can still reply before the author-reviewer discussion period ends. If our response resolves your concerns, we kindly ask you to consider raising the rating of our work. Thank you very much for your time and efforts!

---

> ### Comment · Reviewer_emxE · 2023-11-23
>
> Dear Authors,
>
> Thank you for your comprehensive response.
>
> Regarding W1, your paper addresses the issue of unfairness in text-to-image diffusion models by modifying the text embeddings in the CLIP model. However, it does not address the inherent biases within the diffusion models themselves. Previous studies, such as references [1], [2], [3], and [4], have extensively discussed the biases in the CLIP model. This raises a question about the technical novelty of your approach compared to existing literature. In contrast, addressing the bias of diffusion models themselves is a more worthy area of research, as demonstrated in [4], which tackles biases in both discriminative (like CLIP) and generative models without requiring additional data or training.
>
> [1] Wang J, Liu Y, Wang X E. Are gender-neutral queries really gender-neutral? mitigating gender bias in image search[J]. arXiv preprint arXiv:2109.05433, 2021.
> [2] Berg H, Hall S M, Bhalgat Y, et al. A prompt array keeps the bias away: Debiasing vision-language models with adversarial learning[J]. arXiv preprint arXiv:2203.11933, 2022.
> [3] Seth A, Hemani M, Agarwal C. DeAR: Debiasing Vision-Language Models with Additive Residuals[C]//Proceedings of the IEEE/CVF Conference on Computer Vision and Pattern Recognition. 2023: 6820-6829.
> [4] Chuang C Y, Jampani V, Li Y, et al. Debiasing vision-language models via biased prompts[J]. arXiv preprint arXiv:2302.00070, 2023.
>
> Regarding W2, while I acknowledge the potential of your method to address various forms of unfairness, I disagree with the claim that there is only one prior work ([5]) on fairness in diffusion models. There are other works on this problem, such as those mentioned in references [4], [6], and [7]. A comparative analysis with these works is also missing in your paper, which is a notable omission.
>
> [5] Friedrich F, Schramowski P, Brack M, et al. Fair diffusion: Instructing text-to-image generation models on fairness[J]. arXiv preprint arXiv:2302.10893, 2023.
> [6] Orgad H, Kawar B, Belinkov Y. Editing implicit assumptions in text-to-image diffusion models[J]. arXiv preprint arXiv:2303.08084, 2023.
> [7] Gandikota R, Orgad H, Belinkov Y, et al. Unified concept editing in diffusion models[J]. arXiv preprint arXiv:2308.14761, 2023.
>
> Given these concerns, I maintain my initial rating of the paper.

---

> ### Author Response · Authors · 2023-11-23
> **Official Comment by Authors**
>
> Dear reviewer emxE,
>
> 1. Thank you for your response. Our method solves the problem from a text conditioning perspective, aiming to remove bias generated through diffusion. In comparison, our method as a post-processing method offers advantages such as **Model-Agnostic Integration, Lightweight implementation, and Flexibility.** By employing a post-processing method to remove inherent biases from a view of language bias, as opposed to relying solely on fine-tuning on large-scale ethical datasets, we can reduce efforts of training the whole diffusion model, while still achieving effective bias mitigation. Additionally, our method can be easily applied to existing models, making it scalable and adaptable to various scenarios. It incurs negligible time and space costs while maintaining debiasing effectiveness. This allows for integration with different models and tasks, providing a viable solution. **Therefore, we believe that our approach is innovative and effective.**
>
> 2. Thank you for your advice. The time allocated for our response is limited to 8 hours, and in order to ensure accurate implementation and evaluation, we will consider incorporating the mentioned methods as baselines in future versions if they are currently open source and can be seamlessly implemented on the diffusion model. However, regarding the methods you mentioned about semantic editing, we believe it should not be considered in the context of fairness-related issues. **Even though semantic editing may have the potential for debiasing, they still require the incorporation of agent behavior to achieve fairness. Therefore, a direct comparison should not be made**.
>
> Thank you once again about your comments on our work, if you have any additional questions, please don't hesitate. We are looking forward to your feedback and would greatly appreciate you consider raising the scores. Hope you have a nice day!

---

### Author Response · Authors · 2023-11-19
**For all reviewers: further paper revision**

Dear Area Chairs and reviewers:

Thank you very much to all the reviewers for your comments and suggestions on our work. We greatly appreciate your feedbacks, and we have carefully addressed and responded to each comment. In this paper, **we propose a novel post-processing debiasing method for diffusion models, Fair Mapping, which is model-agnostic, lightweight and flexible for customization**. In experiments, our method effectively removes biases in diffusions model while preserving the quality of the images with little time and space cost. Our method has a positive social impact by fair generation in human-related descriptions.

The revised paper has been uploaded with blue highlighting to indicate modifications. We make the following revisions to the paper:

1. To further elucidate our motivation for mitigating language bias in diffusion models and give a comprehensive understanding of our work, the revised manuscript incorporates enhancements in **Introduction** to provide heightened motivation and significance (refer to Page 2, Section 1 and Page 3, Section2). Additionally, a supplementary section has been appended to expound on the presence of **language bias** in text-to-image diffusions (refer to Page 15, Appendix B).

2. To facilitate readability and comprehension, **we have provided more specific descriptions of the evaluation metrics, including language bias, Human-CLIP, Diversity, and the scale for human preference (refer to Page 18, Section D.2)**. This helps readers better understand and evaluate our research findings. By offering detailed descriptions of these evaluation metrics, readers will have a clearer understanding of the specific indicators and methods used in our research. Moreover, in Appendix D.1 of the revised version, we have listed all the keywords we used in the paper.

3. We discuss a lightweight framework that prioritizes computational efficiency. The framework is compared to the vanilla Stable Diffusion Model. **We add supplementary experiments for highlighting the efficient training process of the framework**, completing the training in just 50 minutes for 150 occupations in the gender attribute on one V100 GPU. It also compares the time required for image generation, demonstrating that the framework performs well and generates images within a reasonable timeframe. Overall, we emphasize the effectiveness of the framework in expediting the model learning process and generating images while preserving sensitive attributes.

4. **In Appendix C on Page 16, we demonstrate our strengths from three aspects**: 1) Model-Agnostic Integration: Fair Mapping seamlessly integrates with diverse text-to-image diffusion models without requiring modifications to their retraining parameters. 2) Lightweight Implementation: As a post-processing approach, Fair Mapping introduces a trainable linear map, ensuring computational efficiency and minimal additional time costs. 3) Flexibility: Fair Mapping's simple training requirements for each keyword allows for easy substitution with alternative prompts or fairness metrics, enhancing adaptability across different scenarios.

5. **We describe the details of the inference stage in the context of Fair Mapping.** It explains how Fair Mapping ensures robustness and addresses possible biases in the generated content. In Appendix C on Page 16, we introduce an additional detector that adapts the user's input prompt to a training prompt with the closest semantic similarity. The similarity distance between the input prompt and each training prompt is calculated using a pre-trained text encoder, and the training prompt with the smallest distance is identified. If the distance falls below a threshold, the input prompt is transformed to match the identified training prompt. Besides, we also highlight the issue of explicit biased information in the input text and the need for the detector to identify the presence of sensitive attributes.

6. **We compare our method with the debiasing method Fair Diffusion[1].** Our goal was to evaluate their performance in terms of alignment and diversity. Through visual comparison, we found that editing operations have a negative impact on image quality, particularly on facial details. Additionally, our method significantly saved time in generating 100 images, taking only 434 seconds, which is 1029 seconds (approximately 70.3\%) less than the Fair Diffusion model on one V100 GPU. Overall, by avoiding manipulative biases and lightweight model structure, our model assessment becomes more reliable and efficient.

Thank you for your valuable feedback. We have carefully considered your comments and addressed them accordingly. If you have any further questions or need clarification, please don't hesitate to reach out to us.

References

[1] Friedrich F, Schramowski P, Brack M, et al. Fair diffusion: Instructing text-to-image generation models on fairness[J]. arXiv preprint arXiv:2302.10893, 2023.

---

### Meta-Review · Area_Chair_zTso · 2023-12-09

**Metareview:**

The paper tackles the issue of biases in text-to-image AI models, where these models often reflect sociocultural biases. It introduces a "Fair Mapping" module that can be added to existing models to make the images they produce less biased. This module is efficient, altering only a few parameters to save on computational resources. It works by ensuring the text's meaning stays consistent while reducing the gap between different groups in the images. The paper also proposes a new way to measure how fair the images are, called FairScore, aiming to keep the image generation unbiased regardless of the text prompt's content.

**Justification For Why Not Higher Score:**

The reviewer's main concerns with the paper involve its approach and novelty in addressing unfairness in text-to-image diffusion models. They point out that the paper focuses on modifying text embeddings in the CLIP model without addressing the inherent biases in the diffusion models themselves, a topic already extensively covered in existing literature. The reviewer also notes an omission in the comparative analysis, highlighting that the paper overlooks several significant prior works on fairness in diffusion models. Additionally, they criticize the paper for not including a broader range of baselines in debiasing diffusion models, arguing that existing methods have proven capable of addressing multiple types of biases simultaneously.

**Justification For Why Not Lower Score:**

N/A

---

### Decision · Program_Chairs · 2024-01-16

Reject